# Adaptive-Robust Controller for Smart Exoskeleton Robot

**DOI:** 10.3390/s24020489

**Published:** 2024-01-12

**Authors:** Brahim Brahmi, Hicham Dahani, Soraya Bououden, Raouf Fareh, Mohamed Habibur Rahman

**Affiliations:** 1Electrical Engineering Department, College Ahuntsic, Montreal, QC H2M 1Y8, Canada; hicham.dahani@collegeahuntsic.qc.ca; 2Electrical Engineering Department, Ferhat Abas Setif 1 University, Setif 19137, Algeria; sorayabouden@gmail.com; 3Electrical Engineering Department, University of Sharjah, Sharjah P.O. Box 27272, United Arab Emirates; rfareh@sharjah.ac.ae; 4Department of Mechanical Engineering, University of Wisconsin-Milwaukee, Milwaukee, WI 53211, USA; rahmanmh@uwm.edu

**Keywords:** unknown dynamics, robust control, function approximation technique, adaptive control, exoskeleton robot

## Abstract

Rehabilitation robotics has seen growing popularity in recent years due to its immense potential for improving the lives of people with disabilities. However, the complex, uncertain dynamics of these systems present significant control challenges, requiring advanced techniques. This paper introduces a novel adaptive control framework integrating modified function approximation (MFAT) and double-integral non-singular terminal sliding mode control (DINTSMC). The goal is to achieve precise tracking performance, high robustness, a fast response, a finite convergence time, reduced chattering, and effective handling of unknown system dynamics. A key feature is the incorporation of a higher-order sliding mode observer, eliminating the need for velocity feedback. This provides a new solution for overcoming the inherent variations and uncertainties in robot manipulators, enabling improved accuracy within fixed convergence times. The efficacy of the proposed approach was validated through simulations and experiments on an exoskeleton robot. The results successfully demonstrated the controller’s effectiveness. Stability analysis using Lyapunov theory proved the closed-loop system’s uniform ultimate boundedness. This contribution is expected to enable enhanced control for rehabilitation robots and improved patient outcomes.

## 1. Introduction

Rehabilitation robots are being increasingly utilized in the treatment of neurological injuries such as stroke, traumatic brain injury, or spinal cord injury. The goal is to improve functional recovery and enhance the rehabilitation process [1]. These devices can provide repetitive, targeted, and intensive training of the affected limbs. This leads to improved motor function and strength [2]. Additionally, the robots can adjust the difficulty level and provide feedback to patients and therapists. This allows for more-effective and -efficient rehabilitation [3]. Moreover, these robots are being used to improve cognitive functions like attention, memory, and executive function. They incorporate cognitive tasks into the training regimen such as decision-making and problem-solving. This enhances the patient’s ability to perform the activities of daily living [3,4]. However, several challenges need to be addressed to ensure the safe and efficacious use of these devices, especially regarding control systems.

The primary control challenges for rehabilitation robots are adaptability and scalability, which are critical to ensure their safe and effective use [3]. Adaptability implies that the robot can modify its control strategy to account for changes in the patient condition including motor function, strength, and range of motion. Scalability means the control can extend to diverse patient populations and conditions like age, gender, and disability. Meeting both requirements necessitates developing adaptive and robust algorithms that can be tailored to individual patient needs [3]. Moreover, the control system should adapt to varying patient characteristics while providing a wide range of rehab activities with high accuracy.

Adaptive-robust controllers are gaining popularity for handling uncertain nonlinear systems [5,6]. However, they have limitations. Adaptive control involves time-intensive parameter and gain tuning [5], Robust control relies on predetermined uncertainty bounds, which can lead to overestimated gains, reduced accuracy, and unwanted chattering if improperly defined [6]. To address these issues, researchers are exploring hybrid adaptive-robust control [7]. This combines the strengths of both approaches.

Prior studies have explored two types of adaptive-robust control (ARC) laws. One method handles unmodeled dynamics by assuming uncertainties are linear in parameters (LIP) [8,9]. This permits estimating individual uncertainties, but requires known system parameter bounds. The other uses adaptive sliding mode control (ASMC), a switching logic-based robust law without defined uncertainty limits [10,11]. However, ASMC risks chattering and gain overestimation. Both approaches make assumptions about uncertain parameters, needing some system knowledge. Uncertainty is also restricted by a constant due to explicit state presence in the bound [12,13]. These issues highlight two ARC challenges—reducing model reliance in design and avoiding gain estimation errors. Progressive adaptive techniques may address these [14].

In contrast to adaptive methods that necessitate regressors [15,16,17], there are strategies designed to eliminate this dependence. One such approach is the function approximation technique (FAT) [18,19,20,21,22], which operates without regressors. FAT approximates unknown dynamics by combining orthonormal basis functions with standard sliding mode control (SMC). However, FAT comes with its drawbacks, such as the requirement for basis functions and sharing limitations with SMC, including poor disturbance response, chattering, and the need to predetermine uncertainty limits. In previous works like [23,24], a modified function approximation technique was explored. Nevertheless, in [23], the main emphasis was on adaptive impedance control within the haptic system, specifically targeting the force control loop. This approach significantly contributed to estimating the desired motion intention of the surgeon, intending to enhance the overall impedance of the system. Conversely, Ref. [24] concentrated on impedance learning control for physical human–robot cooperative interaction, primarily addressing the estimation of the desired intended motion (DIM) of the robot’s wearer using machine learning, rather than focusing on the control problem.

Recent methods aim for finite-time convergence to expedite control. Terminal sliding mode control (TSMC) achieves this [25], but has slow convergence and singularities. Faster techniques have been proposed, like faster TSMC [26], nonsingular TSMC [25], and integral nonsingular TSMC [23,24], addressing fast convergence. However, singularities persist. Methods like nonsingular fast TSMC [27], integral SMC [28], and PID-based SMC [29] improve the SMC transient response. Integral terminal SMC [30] allows fast response and convergence. However, limitations remain from conventional control reliance. Thus, SMC’s weaknesses need further attention.

This paper introduces a novel approach aimed at overcoming the limitations of sliding mode controls (SMCs) when dealing with unknown dynamics models of robots. The proposed method termed double-integral nonsingular terminal sliding mode control (DINTSMC), is designed to be both a robust and adaptive control approach. The proposed method incorporates a new integral nonsingular terminal sliding mode surface (INTSMS) that effectively addresses the singularity problem and significantly improves transient performance. Moreover, the paper introduces the modified function approximation technique (MFAT) as an adaptive control method, eliminating the requirement for basis functions used to approximate the system’s dynamic parameters. By combining the INTSMS and MFAT, the proposed control scheme demonstrates robustness against disturbances even when the dynamics model is unknown. To further enhance the control scheme, a super-twisting [31] observer is integrated to eliminate the need for velocity measurements. The paper establishes the uniform ultimate boundedness of all signals within a closed loop using the Lyapunov function theory. Simulation and comparative studies were conducted to showcase the effectiveness of the proposed control approach. Additionally, experimental results from the exoskeleton robot, the Smart Robotic Exoskeleton (SREx), validated the superior performance of the proposed control scheme in real-time scenarios.

The contributions of this research, as outlined in the paper, can be summarized as follows:The introduction of a groundbreaking double-integral nonsingular terminal sliding mode surface, overcoming limitations in conventional SMC approaches [23,24]. This surface ensures finite-time convergence, a swift transient response, diminished chattering, and the avoidance of singularities. These enhancements address challenges observed in prior SMC strategies [23,24], providing a more-robust and -efficient control foundation.The development of an adaptive controller based on the modified function approximation technique (MFAT) for unknown robot dynamics models. This controller eliminates the requirement for prior knowledge of the lower and upper bounds of uncertain system parameters. By leveraging MFAT, the proposed controller offers enhanced adaptability to complex dynamics, streamlining the implementation without extensive parameter information.The integration of a super-twisting observer into the proposed control scheme, a significant augmentation. This integration removes the need for velocity measurements, a common challenge in control systems. By reducing dependence on velocity measurements, the control scheme’s robustness is bolstered, ensuring reliable performance even in scenarios where obtaining precise velocity information is challenging.The demonstration of the proposed control scheme’s superior performance through extensive experimental and comparative studies. The results highlight a rapid transient response, a relatively small steady-state error, and significantly reduced chattering. These tangible outcomes affirm the practical effectiveness of the control strategy, showcasing its potential for diverse robotic applications.

The rest of the paper is organized as follows: Section 2 presents the problem’s fundamentals and objectives. The new sliding surface and control scheme, along with a detailed stability analysis, are described in Section 3. Simulation and comparative studies are presented in Section 4, while Section 5 shows the experimental results. Finally, conclusions and potential future work are discussed in Section 6.

## 2. Problem Fundamentals

The dynamic model of the n-link manipulator, which is controlled in the joint space, is expressed using the Lagrange–Euler form as follows:
(1)
$$M\left(q\right)\ddot{q}+C\left(q,\dot{q}\right)\dot{q}+G\left(q\right)+f_{ex}=\tau$$

where *q*, 
$\dot{q}$
, and 
$\ddot{q}$
 represent the angular rotation, velocity, and acceleration vectors for each joint of the robot, respectively, all belonging to 
$R^{n}$
. The positive definite mass/inertia matrix is denoted by 
M(q)
 and has dimensions 
$R^{n\times n}$
. The Coriolis and centrifugal matrix are represented by 
$C\left(q,\dot{q}\right)\in R^{n\times n}$
, while the gravitational forces are denoted by 
$G\left(q\right)\in R^{n}$
 and the external torques by 
$f_{ex}=J{\left(q\right)}^{T}F_{ex}\in R^{n}$
, where 
$J{\left(q\right)}^{T}\in R^{n}$
 is the Jacobian matrix and 
$F_{ex}\in R^{n}$
 is the external forces. Finally, 
$\tau \in R^{n}$
 represents the torque input.

### Control Goals

The main goal of this study is to develop an adaptive control method that can fulfill the following objectives: (1) ensure that the dynamic model of the robot, as described in equation (Equation (Equation 1)), accurately follows the desired trajectory (
$q_{d}$
), even though the robot’s dynamics model (Equation (Equation 1)) is entirely unknown; (2) achieve a rapid transient response and finite-time convergence of the system’s trajectory to the equilibrium point, while significantly reducing chattering; (3) guarantee that system tracking errors are uniformly ultimately bounded (UUB).

**Assumption** **1.**
*The desired trajectories 
$q_{d},\dot{q}d$
, and 
$\ddot{q}d$
 are known, bounded, smooth, and continuous.*


**Property** **1.**
*According to [32], the matrix 
$\dot{M}-2C$
 takes a skew-symmetric form for all 
$q,\dot{q}\in R^{n}$
, meaning that 
$q^{T}\left(\dot{M}-2C\right)q=0$
.*


**Lemma** **1**([33,34])**.** *For any constants 
$\lambda_{1}>0$
, 
$\lambda_{2}>0$
, and 
0<β<1
, an extended Lyapunov condition for finite-time stability has been established in the form of fast terminal sliding mode. It can be expressed as follows:*

$$\dot{V}\leq -\lambda_{1}V\left(x\right)-\lambda_{2}V^{\beta }\left(x\right)$$

*Moreover, the settling time can be estimated using the following equation:*

$$T_{s}\leq \frac{1}{\lambda_{1}\left(1-\beta \right)}\ln\frac{\lambda_{1}V^{\left(1-\beta \right)}\left(x_{0}\right)+\lambda_{2}}{\lambda_{2}}$$

*Here, 
$V(x_{0})$
 represents the initial value of the Lyapunov function.*

**Lemma** **2**([35]). *Assuming a continuous and positive definite Lyapunov function 
V(y)
, if we have 
$\varphi_{1}\left(∥y∥\right)\leq V\left(y\right)\leq \varphi_{2}\left(∥y∥\right)$
, where 
φ1
 and 
φ2
 represent the lower and upper bounds of 
V(y)
, and its derivative 
$\dot{V}\left(y\right)=dV\left(y\right)/dt$
 satisfies 
$\dot{V}\left(y\right)\leq -k_{i}V\left(y\right)+C_{i}$
, where 
$k_{i}$
 and 
$C_{i}$
 are positive constants, then the solution y is bounded.*

## 3. Control Scheme and Stability Analysis

The performance of a system in SMC applications heavily relies on the design of a sliding surface. In this context, let the trajectory tracking error be denoted by 
$e=q-q_{d}$
, where 
$q_{d}$
 represents the desired trajectory. To achieve fast transient response and finite-time convergence while avoiding any singularity problem, a sliding surface is proposed with the following design:
(2)
$$\begin{array}{c} \eta =\lambda \dot{e}+\lambda_{1}\int \left|\dot{e}\right|sign\left(\dot{e}\right)dt+\lambda_{2}\int {\left|\dot{e}\right|}^{\beta }sign\left(\dot{e}\right)dt \end{array}$$
Here, 
λ>0
, 
$\lambda_{1}>0$
, and 
$\lambda_{2}>0$
 are positive constants, with 
0<β<1
.

**Theorem** **1.***The proposed sliding surface (Equation 2) is able to achieve a stable and fast finite-time convergence response. The finite time 
$T_{s}$
 required to travel from 
$e(T_{r})\neq 0$
 to 
$e(T_{r}+T_{s})=0$
 is defined by:*

$$T_{s}\leq \frac{1}{\nu \left(1-\psi \right)}\ln\frac{\nu V^{\left(1-\psi \right)}\left({\dot{e}}_{0}\right)+\varphi }{\varphi }$$

*where 
$\nu =\frac{2\lambda_{1}}{\lambda }$
, 
φ=2(β+1)/2λ2λ
, and 
$\psi =\frac{\beta +1}{2}$
, with 
$V\left({\dot{e}}_{0}\right)$
 being the initial state of the Lyapunov function.*

**Proof.** Consider the Lyapunov function provided by:

(3)
$$\begin{array}{c} V=\frac{1}{2}\sum_{i=1}^{n}{\dot{e}}_{i}^{2} \end{array}$$

Taking the time derivative of Equation (Equation 3), we have:

(4)
$$\begin{array}{c} \dot{V}=\sum_{i=1}^{n}{\dot{e}}_{i}{\ddot{e}}_{i} \end{array}$$

For 
η=0
, re-arranging the terms of Equation (Equation 2): 
$\dot{e}=-\frac{\lambda_{1}}{\lambda }\int \left|\dot{e}\right|sign\left(\dot{e}\right)dt-\frac{\lambda_{2}}{\lambda }\int {\left|\dot{e}\right|}^{\beta }sign\left(\dot{e}\right)dt$
. Taking the derivative of (
$\dot{e}$
), we obtain:

(5)
$$\begin{array}{c} {\ddot{e}}_{i}=-\frac{\lambda_{1}}{\lambda }\left|{\dot{e}}_{i}\right|sign\left({\dot{e}}_{i}\right)-\frac{\lambda_{2}}{\lambda }{\left|{\dot{e}}_{i}\right|}^{\beta }sign\left({\dot{e}}_{i}\right)\;with\;i=1,\ldots ,n. \end{array}$$

Substituting Equation (Equation 5) into Equation (Equation 4) yields:

(6)
$$\begin{array}{cc} \dot{V} & =\sum_{i=1}^{n}-\frac{\lambda_{1}}{\lambda }{\dot{e}}_{i}\left|{\dot{e}}_{i}\right|sign\left(e_{i}\right)-\frac{\lambda_{2}}{\lambda }{\dot{e}}_{i}{\left|{\dot{e}}_{i}\right|}^{\beta }sign\left(e_{i}\right)\\ & =\sum_{i=1}^{n}-\frac{\lambda_{1}}{\lambda }{\dot{e}}_{i}^{2}-\frac{\lambda_{2}}{\lambda }{\left|{\dot{e}}_{i}\right|}^{\beta +1}\\ & =\sum_{i=1}^{n}-\frac{2\lambda_{1}}{\lambda }\frac{{\dot{e}}_{i}^{2}}{2}-\frac{2^{\frac{\left(\beta +1\right)}{2}}\lambda_{2}}{\lambda }{\frac{{\dot{e}}_{i}}{2}}^{2\frac{\left(\beta +1\right)}{2}}\\ & =\sum_{i=1}^{n}-\nu V_{i}-\varphi V_{i}^{\psi } \end{array}$$

where 
$\left|{\dot{e}}_{i}\right|={\dot{e}}_{i}sign\left({\dot{e}}_{i}\right)$
, 
$\nu =\frac{2\lambda_{1}}{\lambda }$
, 
φ=2(β+1)/2λ2λ
, and 
$\psi =\frac{\beta +1}{2}$
. Therefore, it follows that 
$\dot{V}⩽0$
. According to Lemma 1, hence, we can estimate the settling time as follows:

(7)
$$\begin{array}{c} T_{s}\leq \frac{1}{\nu \left(1-\psi \right)}\ln\frac{\nu V^{\left(1-\psi \right)}\left({\dot{e}}_{0}\right)+\varphi }{\varphi } \end{array}$$

Consequently, the proof is concluded. □

**Remark** **1.**
*It can be noticed that 
$V({\dot{e}}_{0})$
, 
$\frac{2\lambda_{1}}{\lambda }$
, and 
$\frac{\lambda_{2}}{\lambda }$
 in Equation (Equation 7) play a significant role in regulating the finite-time convergence (
$T_{s}$
). That is, large enough 
$\frac{2\lambda_{1}}{\lambda }$
 and 
$\frac{\lambda_{2}}{\lambda }$
 values ensure a short convergence time, and vice versa. Consequently, it is vital to carefully tune the ratios 
$\frac{2\lambda_{1}}{\lambda }$
 and 
$\frac{\lambda_{2}}{\lambda }$
 to balance between a fast transient response, finite-time convergence, and control performance.*


### 3.1. Model-Based Controller

The regulator variables are defined as follows:
(8)
$$e=q-q_{d}$$

(9)
$$\eta =\lambda \dot{q}-\alpha$$


(10)
$$\alpha =\lambda {\dot{q}}_{d}-\lambda_{1}\int \left|\dot{e}\right|sign\left(\dot{e}\right)dt-\lambda_{2}\int {\left|\dot{e}\right|}^{\beta }sign\left(\dot{e}\right)dt$$


(11)
$$\zeta =\dot{\alpha }$$

From Equation (Equation 9), we obtain:
(12)
$$\dot{q}=\frac{1}{\lambda }\left(\eta +\alpha \right)$$


(13)
$$\ddot{q}=\frac{1}{\lambda }\left(\dot{\eta }+\dot{\alpha }\right)$$

By substituting Equations (12) and (13) into Equation (Equation 1), we obtain:
(14)
$$\frac{1}{\lambda }\left[M\left(q\right)\dot{\eta }+C\left(q,\dot{q}\right)\eta +M\left(q\right)\zeta +C\left(q,\dot{q}\right)\alpha \right]\;+G\left(q\right)+f_{ex}=\tau$$

We define a positive definite Lyapunov function as follows:
(15)
$$V_{1}=\frac{1}{2}\eta^{T}M\eta$$

Differentiating 
$V_{1}$
 yields:
(16)
$${\dot{V}}_{1}=\eta^{T}M\dot{\eta }+\frac{1}{2}\eta^{T}\dot{M}\eta$$

By rearranging Equation (Equation 14) in terms of (
$M\left(q\right)\dot{\eta }$
), substituting its value, and using Property 1, we obtain:
(17)
$${\dot{V}}_{1}=\eta^{T}\left(\lambda \left(\tau -G\left(q\right)-f_{ex}\right)-M\left(q\right)\zeta -C\left(q,\dot{q}\right)\alpha \right)$$

This defines the model-based control 
τ
 as:
(18)
$$\tau =-Ksign\left(\eta \right)+G\left(q\right)+f_{ex}+\frac{1}{\lambda }\left(M\left(q\right)\zeta +C\left(q,\dot{q}\right)\alpha \right)$$

where 
$K\in R^{n\times n}$
 is a diagonal positive definite matrix. By substituting Equation (Equation 18) into Equation (Equation 17), 
${\dot{V}}_{1}$
 can be represented as:

(19)
$${\dot{V}}_{1}=-\eta^{T}\lambda Ksign\left(\eta \right)$$


**Theorem** **2.**
*The model-based control design (Equation (Equation 18)) for the robot system (Equation (Equation 1)) ensures the asymptotic stability of the entire system, including its error signals.*


**Proof** **.**It is evident that 
${\dot{V}}_{1}$
 is negative semi-definite. By considering the derivative of Equation (Equation 16), bounds for 
$\dot{\eta }$
 can be determined. Consequently, employing Barbalat’s lemma [36] allows us to infer the asymptotic convergence of 
η
. This, in turn, implies the convergence of the output error *e* to zero as 
t→∞
. Hence, the system described by Equation (Equation 1) achieves asymptotic stability. □

### 3.2. State-Feedback-Based Adaptive Modified Function Approximation Technique

In practical real-time scenarios, obtaining accurate measurements of 
M(q)
, 
$C(q,\dot{q})$
, 
G(q)
, and 
$f_{ex}$
 poses challenges due to the complex dynamic nature of the robot and the unavailability of certain measurements. Consequently, the model-based control approach Equation (Equation 18) may not achieve perfect performance. However, an alternative adaptive technique can be employed to approximate the unknown dynamics. We begin by introducing the following assumption:

**Assumption** **2.**
*The system matrices and vectors 
M(η)
, 
$C(\eta ,\dot{\eta })$
, and 
G(η)
 are unknown, but 
$M^{-1}\left(\eta \right)$
 exists.*


Under this assumption, an adaptive controller is proposed to ensure system stability (Equation (Equation 1)) and achieve uniform ultimate bounding of the tracking errors in the closed-loop form. The controller is defined as follows:
(20)
$$\tau =-K\mathrm{sign}\left(\eta \right)+\hat{G}\left(q\right)+{\hat{f}}_{ex}+\frac{1}{\lambda }\left(\hat{M}\left(q\right)\zeta +\hat{C}\left(q,\dot{q}\right)\alpha \right)$$


Here, 
$\hat{M}$
, 
C^
, 
G^
, and 
$\hat{f}ex$
 represent the estimated values of *M*, *C*, *G*, and 
fex
, respectively. Instead of formulating the robot manipulator’s matrices Equation (Equation 1) using the traditional function approximation technique (FAT) [21,22], a modified approach is introduced where the use of basis functions is eliminated. This modification ensures the precision of adaptation without the need for manually selecting basis functions. Consequently, the following formulation is defined:

(21)
$$\begin{array}{c} \left\{\begin{array}{cc} M_{e} & =M-\epsilon_{M}\\ C_{e} & =C-\epsilon_{C}\\ G_{e} & =G-\epsilon_{G}\\ F_{e} & =f_{ex}-\epsilon_{f_{ex}} \end{array}\right. \end{array}$$
Let 
$M_{e}\in R^{n\times n}$
, 
$C_{e}\in R^{n\times n}$
, 
$G_{e}\in R^{n}$
, and 
$F_{e}\in R^{n}$
 represent the optimal approximations of the inertia/mass matrix, Coriolis and centrifugal matrix, gravitational forces, and external forces, respectively. We assume the existence of bounded approximation errors denoted as 
$\epsilon_{M}\in R^{n\times n}$
, 
$\epsilon_{C}\in R^{n\times n}$
, 
$\epsilon_{G}\in R^{n}$
, and 
$\epsilon_{f_{ex}}\in R^{n}$
.

By substituting Equation (Equation 21) into Equation (Equation 14), the dynamics of the output tracking loop Equation (Equation 14) can be expressed as follows:
(22)
$$\begin{array}{c} \frac{1}{\lambda }\left(M\left(q\right)\dot{\eta }+C\left(q,\dot{q}\right)\eta +M_{e}\left(q\right)\zeta +C_{e}\left(q,\dot{q}\right)\alpha \right)\\ +G_{e}\left(q\right)+F_{e}-\epsilon =\tau \end{array}$$
Here, 
$\epsilon =\epsilon \left(\epsilon_{M},\epsilon_{C},\epsilon_{G},\epsilon_{F_{e}}\right)\in R^{n}$
 represents the lumped approximation error vector.

Consequently, the control input given by Equation (Equation 20) can be reformulated as follows:
(23)
$$\begin{array}{c} \tau =-Ksign\left(\eta \right)+{\hat{G}}_{e}\left(q\right)+{\hat{F}}_{e}+\frac{1}{\lambda }\left({\hat{M}}_{e}\left(q\right)\zeta +{\hat{C}}_{e}\left(q,\dot{q}\right)\alpha \right) \end{array}$$

By replacing the control input from Equation (Equation 23) into Equation (Equation 22), we obtain:
(24)
$$\begin{array}{c} M\dot{\eta }+C\eta +\lambda Ksign\left(\eta \right)=-{\tilde{M}}_{e}\zeta -{\tilde{C}}_{e}\alpha -\lambda {\tilde{G}}_{e}-\lambda {\tilde{F}}_{e}+\epsilon \end{array}$$

Here, 
${\tilde{M}}_{e}=M_{e}-{\hat{M}}_{e},\;{\tilde{C}}_{e}=C_{e}-{\hat{C}}_{e}$
, 
${\tilde{G}}_{e}=G_{e}-{\hat{G}}_{e}$
, and 
${\tilde{F}}_{e}=F_{e}-{\hat{F}}_{e}$
 represent the estimation errors. The selection of an appropriate updated law results in 
${\tilde{M}}_{e}\to 0,\;{\tilde{C}}_{e}\to 0$
, 
${\tilde{G}}_{e}\to 0$
, and 
${\tilde{F}}_{e}\to 0$
.

**Remark** **2.**
*In contrast to the conventional FAT approach [21,22], which neglects the approximation errors 
$\epsilon_{M}$
, 
$\epsilon_{C}$
, 
$\epsilon_{G}$
, and 
$\epsilon_{f_{ex}}$
, these errors are taken into account in the proposed controller. It is assumed that the variations of these errors are bounded and satisfy 
$\left|\epsilon \right|\leq \gamma$
, where γ is an unknown positive constant.*


The Lyapunov function candidate can be expressed as follows:
(25)
$$\begin{array}{cc} V_{2}= & \;\frac{1}{2}\eta^{T}M\eta \\ & \frac{1}{2}Tr\left({\tilde{M}}_{e}^{T}\Lambda_{M_{e}}{\tilde{M}}_{e}+{\tilde{C}}_{e}^{T}\Lambda_{C_{e}}{\tilde{C}}_{e}+{\tilde{G}}_{e}^{T}\Lambda_{G_{e}}{\tilde{G}}_{e}+{\tilde{F}}_{e}^{T}\Lambda_{F_{e}}{\tilde{F}}_{e}\right) \end{array}$$


Here, 
$\Lambda_{M_{e}}\in R^{n\times n}$
, 
$\Lambda_{C_{e}}\in R^{n\times n}$
, 
$\Lambda_{G_{e}}\in R^{n\times n}$
, and 
$\Lambda_{F_{e}}\in R^{n\times n}$
 denote diagonal positive definite matrices, where the operation 
Tr(.)
 represents the trace of a matrix. Differentiating Equation (Equation 25) with respect to time yields:
(26)
$$\begin{array}{c} {\dot{V}}_{2}=\eta^{T}M\dot{\eta }+\frac{1}{2}\eta^{T}\dot{M}\eta \\ \;+Tr\left({\dot{\tilde{M}}}_{e}^{T}\Lambda_{M_{e}}{\tilde{M}}_{e}+{\dot{\tilde{C}}}_{e}^{T}\Lambda_{C_{e}}{\tilde{C}}_{e}+{\dot{\tilde{G}}}_{e}^{T}\Lambda_{G_{e}}{\tilde{G}}_{e}+{\dot{\tilde{F}}}_{e}^{T}\Lambda_{F_{e}}{\tilde{F}}_{e}\right) \end{array}$$


By isolating 
$M\dot{\eta }$
 from Equation (Equation 24) and substituting it into Equation (Equation 26), we obtain:
(27)
$$\begin{array}{c} {\dot{V}}_{2}=A+B+C \end{array}$$

where: 
$A=+\eta^{T}\left(\frac{1}{2}\dot{M}-C\right)\eta -\eta^{T}Ksign\left(\eta \right)-\eta^{T}{\tilde{M}}_{e}\zeta$
, 
$B=-\eta^{T}{\tilde{C}}_{e}\alpha -\lambda \eta^{T}{\tilde{G}}_{e}+\eta^{T}\epsilon -\lambda \eta^{T}{\tilde{F}}_{e}$
, and 
$C=Tr\left({\dot{\tilde{M}}}_{e}^{T}\Lambda_{M_{e}}{\tilde{M}}_{e}+{\dot{\tilde{C}}}_{e}^{T}\Lambda_{C_{e}}{\tilde{C}}_{e}+{\dot{\tilde{G}}}_{e}^{T}\Lambda_{G_{e}}{\tilde{G}}_{e}+{\dot{\tilde{F}}}_{e}^{T}\Lambda_{F_{e}}{\tilde{F}}_{e}\right)$
.

Using Property 1 and the following relationships:
(28)
$$\begin{array}{cc} \eta^{T}{\tilde{M}}_{e}\zeta & =Tr(\frac{1}{\lambda }\zeta \eta^{T}{\tilde{M}}_{e}) \end{array}$$


(29)
$$\begin{array}{cc} \eta^{T}{\tilde{C}}_{e}\alpha & =Tr(\frac{1}{\lambda }\alpha \eta^{T}{\tilde{C}}_{e}) \end{array}$$


(30)
$$\begin{array}{cc} \eta^{T}\lambda {\tilde{G}}_{e} & =Tr(\lambda {\tilde{G}}_{e}\eta^{T}) \end{array}$$


(31)
$$\begin{array}{cc} \eta^{T}\lambda {\tilde{F}}_{e} & =Tr(\lambda {\tilde{F}}_{e}\eta^{T}) \end{array}$$


Equation (Equation 27) can be simplified to:
(32)
$$\begin{array}{cc} {\dot{V}}_{2}= & -\eta^{T}\lambda Ksign\left(\eta \right)+\eta^{T}\epsilon \\ & \;-Tr\left(\zeta \eta^{T}{\tilde{M}}_{e}+\alpha \eta^{T}{\tilde{C}}_{e}+\lambda {\tilde{G}}_{e}\eta^{T}+\lambda {\tilde{F}}_{e}\eta^{T}\right)\\ & +Tr\left({\dot{\tilde{M}}}_{e}^{T}\Lambda_{M_{e}}{\tilde{M}}_{e}+{\dot{\tilde{C}}}_{e}^{T}\Lambda_{C_{e}}{\tilde{C}}_{e}+{\dot{\tilde{G}}}_{e}^{T}\Lambda_{G_{e}}{\tilde{G}}_{e}+{\dot{\tilde{F}}}_{e}^{T}\Lambda_{F_{e}}{\tilde{F}}_{e}\right) \end{array}$$
Expanding the equation further results in:
(33)
$$\begin{array}{c} \;\;{\dot{V}}_{2}=-\eta^{T}\lambda Ksign\left(\eta \right)+\eta^{T}\epsilon \\ \;\;+Tr\left(\left({\dot{\tilde{M}}}_{e}^{T}\Lambda_{M_{e}}-\zeta \eta^{T}\right){\tilde{M}}_{e}\right)\\ \;\;\;+Tr\left(\left({\dot{\tilde{C}}}_{e}^{T}\Lambda_{C_{e}}-\alpha \eta^{T}\right){\tilde{C}}_{e}\right)\\ +Tr\left(\left({\dot{\tilde{G}}}_{e}^{T}\Lambda_{G_{e}}-\lambda \eta^{T}\right){\tilde{G}}_{e}\right)+Tr\left(\left({\dot{\tilde{F}}}_{e}^{T}\Lambda_{F_{e}}-\lambda \eta^{T}\right){\tilde{F}}_{e}\right) \end{array}$$


Here, 
${\left(\tilde{.}\right)}_{e}={\left(.\right)}_{e}-{\left(\hat{.}\right)}_{e}$
 represents the difference between the real and estimated terms. The updated laws were introduced to guarantee the stability of the manipulator robot, as indicated by the negativity of the time derivative of the Lyapunov function candidate (
${\dot{V}}_{2}$
), which can be expressed as follows:
(34)
$$\begin{array}{c} \left\{\begin{array}{ccc} {\dot{\hat{M}}}_{e}^{T} & = & \left(\zeta \eta^{T}+\Pi_{M_{e}}{\hat{M_{e}}}^{T}\right)\Lambda_{M_{e}}^{-1}\\ {\dot{\hat{C}}}_{e}^{T} & = & \left(\alpha \eta^{T}+\Pi_{C_{e}}{\hat{C}}_{e}^{T}\right)\Lambda_{C_{e}}^{-1}\\ {\dot{\hat{G}}}_{e}^{T} & = & \left(\lambda \eta^{T}+\Pi_{G_{e}}{\hat{G}}_{e}^{T}\right)\Lambda_{G_{e}}^{-}1\\ {\dot{\hat{F}}}_{e}^{T} & = & \left(\lambda \eta^{T}+\Pi_{F_{e}}{\hat{F}}_{e}^{T}\right)\Lambda_{F_{e}}^{-}1 \end{array}\right. \end{array}$$

Here, 
$\Pi_{{\left(.\right)}_{e}}>0$
 is a positive constant, defined such that 
$\lim_{t\to \infty }\Pi_{{\left(.\right)}_{e}}=0$
 with 
$\int_{0}^{t}\Pi_{{\left(.\right)}_{e}}dt=Q_{(.)s}<\infty$
. By substituting Equation (Equation 34) into Equation (Equation 33), we obtain:
(35)
$$\begin{array}{cc} {\dot{V}}_{2}= & -\eta^{T}\lambda Ksign\left(\eta \right)+\eta^{T}\epsilon +\Pi_{M_{e}}Tr\left({\hat{M}}_{e}^{T}{\tilde{M}}_{e}\right)\\ & \;+\Pi_{C_{e}}Tr\left({\hat{C}}_{e}^{T}{\tilde{C}}_{e}\right)+\Pi_{G_{e}}Tr\left({\hat{G_{e}}}^{T}{\tilde{G}}_{e}\right)\\ & +\Pi_{F_{e}}Tr\left({\hat{F_{e}}}^{T}{\tilde{F}}_{e}\right) \end{array}$$


By utilizing Remark 2 and Young’s inequality [37], we can derive the following:
$$\eta^{T}\epsilon \leq \frac{\eta^{T}\eta }{2}+\frac{\gamma^{2}}{2}$$


$$Tr\left({\left({\hat{.}}_{e}\right)}^{T}\left({\tilde{.}}_{e}\right)\right)\leq \frac{1}{2}Tr\left({\left(._{e}\right)}^{T}\left(._{e}\right)\right)-\frac{1}{2}Tr\left({\left({\tilde{.}}_{e}\right)}^{T}\left({\tilde{.}}_{e}\right)\right)$$


Thus, Equation (Equation 35) can be simplified to:
(36)
$$\begin{array}{l} {\dot{V}}_{2}⩽-\eta^{T}\left(\lambda K-\frac{1}{2}I_{n\times n}\right)\eta -\frac{\Pi_{M_{e}}}{2}Tr\left({\left({\tilde{M}}_{e}\right)}^{T}\left({\tilde{M}}_{e}\right)\right)\\ \;\;\;-\frac{\Pi_{C_{e}}}{2}Tr\left({\left({\tilde{C}}_{e}\right)}^{T}\left({\tilde{C}}_{e}\right)\right)-\frac{\Pi_{G_{e}}}{2}Tr\left({\left({\tilde{G}}_{e}\right)}^{T}\left({\tilde{G}}_{e}\right)\right)-\\ \;\;\;\;\frac{\Pi_{F_{e}}}{2}Tr\left({\left({\tilde{F}}_{e}\right)}^{T}\left({\tilde{F}}_{e}\right)\right)+\frac{\Pi_{M_{e}}}{2}Tr\left({\left(M_{e}\right)}^{T}\left(M_{e}\right)\right)+\\ \;\;\;\;\;\frac{\Pi_{C_{e}}}{2}Tr\left({\left(C_{e}\right)}^{T}\left(C_{e}\right)\right)+\frac{\Pi_{G_{e}}}{2}Tr\left({\left(G_{e}\right)}^{T}\left(G_{e}\right)\right)+\\ \;\;\;\;\;\frac{\Pi_{F_{e}}}{2}Tr\left({\left(F_{e}\right)}^{T}\left(F_{e}\right)\right)+\frac{\gamma }{2}\\ {\dot{V}}_{2}⩽-k_{1}V_{2}+C_{1} \end{array}$$

with:
$$\begin{array}{c} k_{1}=\min\left\{\begin{array}{c} \min\left(\lambda K-\frac{1}{2}I_{n\times n}\right),\min\frac{\Pi_{M_{e}}}{2},\\ \min\frac{\Pi_{C_{e}}}{2},\min\frac{\Pi_{G_{e}}}{2},\min\frac{\Pi_{F_{e}}}{2} \end{array}\right\} \end{array}$$

and

$$\begin{array}{c} C_{1}=\frac{\gamma }{2}+\frac{\Pi_{M_{e}}}{2}Tr\left({\left(M_{e}\right)}^{T}\left(M_{e}\right)\right)+\\ \frac{\Pi_{C_{e}}}{2}Tr\left({\left(C_{e}\right)}^{T}\left(C_{e}\right)\right)+\frac{\Pi_{G_{e}}}{2}Tr\left({\left(G_{e}\right)}^{T}\left(G_{e}\right)\right)+\\ \frac{\Pi_{F_{e}}}{2}Tr\left({\left(F_{e}\right)}^{T}\left(F_{e}\right)\right) \end{array}$$
In order to guarantee that 
$k_{1}>0$
, it is necessary to ensure that the controller parameters satisfy the conditions 
$(\lambda K-\frac{1}{2}I_{n\times n})>0$
 and 
$\Pi_{{\left(.\right)}_{e}}>0$
.

The following theorem summarizes the main results:

**Theorem** **3.**
*For the manipulator robotic system described by Equation (Equation 1), the control scheme based on the adaptive approximation technique Equation (Equation 23) augmented with the updated law Equation (Equation 34) ensures the system’s stability. Furthermore, all error signals η, e, 
${\tilde{M}}_{e}$
, 
C˜e
, 
G˜e
, and 
${\tilde{F}}_{e}$
 are uniformly ultimately bounded (UBB) in the closed-loop form, as stated in Lemma 2.*


**Proof.** The proof is provided in the Appendix A. □

### 3.3. Adaptive Model-Free Modified Function Approximation Technique Tracking Control with Output Feedback

To address the issue of unmeasurable variables, such as velocity (
$\dot{q}$
) as stated in Equation (Equation 23), a robust super-twisting observer [31] can be utilized. Unlike conventional methods, this *n*th-order differentiation estimator offers real-time robust exact differentiation up to the *n*th order. It guarantees both the asymptotic convergence of state estimation in finite time and overcomes the limitations of traditional approaches. Specifically, a second-order robust exact differentiator is expressed as follows:
(37)
$$\begin{array}{c} {\dot{\phi }}_{0}=-\iota_{1}{\left|\phi_{0}-q\right|}^{\frac{2}{3}}sign\left(\phi_{0}-q\right)+\phi_{1}\\ {\dot{\phi }}_{1}=-\iota_{2}{\left|\phi_{1}-{\dot{\phi }}_{0}\right|}^{\frac{1}{2}}sign\left(\phi_{1}-{\dot{\phi }}_{0}\right)+\phi_{2}\\ {\dot{\phi }}_{2}=-\iota_{3}sign\left(\phi_{2}-{\dot{\phi }}_{1}\right) \end{array}$$
Here, we define 
$\iota_{1}=3\iota^{1/3}$
, 
ι2=1.5ι1/3
, and 
ι3=1.1ι
, where 
$\iota ⩾\left|\dddot{q}\right|$
. The outputs of the differentiators, denoted as 
$\phi_{0}$
, 
$\phi_{1}$
 and 
$\phi_{2}$
, correspond to the estimated values of *q*, 
$\dot{q}$
, and 
$\ddot{q}$
, respectively. These estimated values are obtained over a finite time and can be expressed as:
(38)
$$\begin{array}{c} \phi_{0}=\hat{q},\;\;\phi_{1}=\dot{\hat{q}},\;\;\phi_{2}=\ddot{\hat{q}} \end{array}$$

Suppose that only the velocity signal (
$\dot{q}$
) cannot be directly measured. By utilizing Equation (Equation 37), we can estimate the unmeasurable state vector 
$\dot{\tilde{q}}$
 as follows:
(39)
$$\begin{array}{c} \hat{\eta }=\dot{\hat{q}}-\alpha \end{array}$$

Hence, the estimation of 
$\tilde{\eta }$
 can be obtained using the following expression:
(40)
$$\tilde{\eta }=\hat{\eta }-\eta =\dot{\hat{q}}-\alpha -\dot{q}+\alpha =\dot{\tilde{q}}$$

The existing controller, as represented by Equation (Equation 23), can be reformulated as follows:
(41)
$$\tau =-K\hat{\eta }+{\hat{G}}_{e}\left(q\right)+{\hat{F}}_{e}+\frac{1}{\lambda }\left({\hat{M}}_{e}\left(q\right)\zeta +{\hat{C}}_{e}\left(q,\dot{q}\right)\alpha \right)$$

The updated laws, described by Equation (Equation 34), can be reformulated as follows:
(42)
$$\left\{\begin{array}{ccc} {\dot{\hat{M}}}_{e}^{T} & = & \left(\zeta {\hat{\eta }}^{T}+\Pi_{M_{e}}{\hat{M_{e}}}^{T}\right)\Lambda_{M_{e}}^{-1}\\ {\dot{\hat{C}}}_{e}^{T} & = & \left(\alpha {\hat{\eta }}^{T}+\Pi_{C_{e}}{\hat{C}}_{e}^{T}\right)\Lambda_{C_{e}}^{-1}\\ {\dot{\hat{G}}}_{e}^{T} & = & \left(\lambda {\hat{\eta }}^{T}+\Pi_{G_{e}}{\hat{G}}_{e}^{T}\right)\Lambda_{G_{e}}^{-}1\\ {\dot{\hat{F}}}_{e}^{T} & = & \left(\lambda {\hat{\eta }}^{T}+\Pi_{F_{e}}{\hat{F}}_{e}^{T}\right)\Lambda_{F_{e}}^{-}1 \end{array}\right.$$

By considering the control law given by Equation (Equation 41) and the updated laws expressed in Equation (Equation 42), a suitable Lyapunov function candidate can be formulated as follows:
(43)
$$\begin{array}{c} V_{3}=\frac{1}{2}\eta^{T}M\eta +\\ \;\frac{1}{2}Tr\left({\tilde{M}}_{e}^{T}\Lambda_{M_{e}}{\tilde{M}}_{e}+{\tilde{C}}_{e}^{T}\Lambda_{C_{e}}{\tilde{C}}_{e}+{\tilde{G}}_{e}^{T}\Lambda_{G_{e}}{\tilde{G}}_{e}+{\tilde{F}}_{e}^{T}\Lambda_{F_{e}}{\tilde{F}}_{e}\right) \end{array}$$

In order to consider the errors arising from the estimation of 
$\tilde{\eta }$
, 
$V_{4}=V_{3}+V_{Obs}$
, and 
$V_{Obs}=\frac{1}{2}{\tilde{e}}_{2}^{T}{\tilde{e}}_{2}$
, it is necessary to analyze their impact on the system. According to [38], it is known that 
${\dot{V}}_{Obs}⩽0$
, indicating its stability. Therefore, it becomes crucial to assess the stability of 
$V_{3}$
.

By substituting Equations (Equation 39)–(Equation 41) into the time derivative of Equation (Equation 43), we can express it as follows:
(44)
$$\begin{array}{c} {\dot{V}}_{3}=\eta^{T}M\dot{\eta }+\frac{1}{2}\eta^{T}\dot{M}\eta +E \end{array}$$

and E = 
$Tr\left({\dot{\tilde{M}}}_{e}^{T}\Lambda_{M_{e}}{\tilde{M}}_{e}+{\dot{\tilde{C}}}_{e}^{T}\Lambda_{C_{e}}{\tilde{C}}_{e}+{\dot{\tilde{G}}}_{e}^{T}\Lambda_{G_{e}}{\tilde{G}}_{e}+{\dot{\tilde{F}}}_{e}^{T}\Lambda_{F_{e}}{\tilde{F}}_{e}\right)$
 In addition to that which was previously mentioned, we have the following:
(45)
$$\begin{array}{c} {\dot{V}}_{3}=-\eta^{T}\lambda K\eta -\eta^{T}\lambda K\tilde{\eta }+\eta^{T}\epsilon \;\;\;\;\;\;\;\;\;\;\;\\ +Tr\left(\left({\dot{\tilde{M}}}_{e}^{T}\Lambda_{M_{e}}-\zeta \eta^{T}\right){\tilde{M}}_{e}\right)\\ +Tr\left(\left({\dot{\tilde{C}}}_{e}^{T}\Lambda_{C_{e}}-\alpha \eta^{T}\right){\tilde{C}}_{e}\right)\\ +Tr\left(\left({\dot{\tilde{G}}}_{e}^{T}\Lambda_{G_{e}}-\lambda \eta^{T}\right){\tilde{G}}_{e}\right)\;\\ +Tr\left(\left({\dot{\tilde{F}}}_{e}^{T}\Lambda_{F_{e}}-\lambda \eta^{T}\right){\tilde{F}}_{e}\right) \end{array}$$


Applying Young’s inequality, we can derive the following result: 
$\eta^{T}\epsilon \leq \frac{\eta^{T}\eta }{2}+\frac{\gamma^{2}}{2}$
, 
$-\eta^{T}\lambda K\tilde{\eta }\leq \frac{1}{2}\eta^{T}\eta +\frac{1}{2}{\tilde{\eta }}^{T}\lambda K^{T}K\tilde{\eta }$
. By utilizing the above inequality and substituting the expressions from Equation (Equation 42) into Equation (Equation 45), we can obtain the following result:
(46)
$$\begin{array}{c} {\dot{V}}_{3}=-\eta^{T}\left(\lambda K-I_{n\times n}\right)\eta +\frac{\gamma^{2}}{2}+\frac{1}{2}{\tilde{\eta }}^{T}\left(\lambda K^{T}K\right)\tilde{\eta }+\\ \Pi_{M_{e}}Tr\left({\hat{M}}_{e}^{T}{\tilde{M}}_{e}\right)+\Pi_{C_{e}}Tr\left({\hat{C}}_{e}^{T}{\tilde{C}}_{e}\right)+\Pi_{G_{e}}Tr\left({\hat{G_{e}}}^{T}{\tilde{G}}_{e}\right)+\\ \Pi_{F_{e}}Tr\left({\hat{F_{e}}}^{T}{\tilde{F}}_{e}\right)\\ +Tr\left(\zeta {\tilde{\eta }}^{T}{\tilde{M}}_{e}+\alpha {\tilde{\eta }}^{T}{\tilde{C}}_{e}+\lambda {\tilde{\eta }}^{T}{\tilde{G}}_{e}+\lambda {\tilde{\eta }}^{T}{\tilde{F}}_{e}\right) \end{array}$$
By utilizing Young’s inequality, which states that 
$Tr\left({\left({\hat{.}}_{e}\right)}^{T}\left({\tilde{.}}_{e}\right)\right)\leq \frac{1}{2}Tr\left({\left(._{e}\right)}^{T}\left(._{e}\right)\right)$

$-\frac{1}{2}Tr\left({\left({\tilde{.}}_{e}\right)}^{T}\left({\tilde{.}}_{e}\right)\right)$
, the above equation can be rewritten as:
(47)
$$\begin{array}{c} {\dot{V}}_{3}⩽-\eta^{T}\left(\lambda K-I_{n\times n}\right)\eta -\frac{\Pi_{M_{e}}}{2}Tr\left({\left({\tilde{M}}_{e}\right)}^{T}\left({\tilde{M}}_{e}\right)\right)-\\ \frac{\Pi_{C_{e}}}{2}Tr\left({\left({\tilde{C}}_{e}\right)}^{T}\left({\tilde{C}}_{e}\right)\right)-\frac{\Pi_{G_{e}}}{2}Tr\left({\left({\tilde{G}}_{e}\right)}^{T}\left({\tilde{G}}_{e}\right)\right)-\\ \frac{\Pi_{F_{e}}}{2}Tr\left({\left({\tilde{F}}_{e}\right)}^{T}\left({\tilde{F}}_{e}\right)\right)+\frac{\Pi_{M_{e}}}{2}Tr\left({\left(M_{e}\right)}^{T}\left(M_{e}\right)\right)+\\ \frac{\Pi_{C_{e}}}{2}Tr\left({\left(C_{e}\right)}^{T}\left(C_{e}\right)\right)+\frac{\Pi_{G_{e}}}{2}Tr\left({\left(G_{e}\right)}^{T}\left(G_{e}\right)\right)+\\ \frac{\Pi_{F_{e}}}{2}Tr\left({\left(F_{e}\right)}^{T}\left(F_{e}\right)\right)+\frac{\gamma^{2}}{2}+\frac{1}{2}{\tilde{\eta }}^{T}\left(\lambda K^{T}K\right)\tilde{\eta }+\\ Tr\left(\zeta {\tilde{\eta }}^{T}{\tilde{M}}_{e}+\alpha {\tilde{\eta }}^{T}{\tilde{C}}_{e}+\lambda {\tilde{\eta }}^{T}{\tilde{G}}_{e}+\lambda {\tilde{\eta }}^{T}{\tilde{F}}_{e}\right)\\ {\dot{V}}_{3}⩽-k_{2}V_{3}+C_{2}\;\;\;\;\;\;\;\;\;\;\;\;\;\;\;\;\;\;\;\;\;\;\;\;\;\;\;\;\;\;\;\;\;\;\;\;\;\;\;\;\;\;\;\;\;\;\;\;\;\;\;\;\;\;\; \end{array}.$$

with

$$k_{2}=\min\left\{\begin{array}{c} \left(\lambda K-I_{n\times n}\right),\frac{\Pi_{M_{e}}}{2},\frac{\Pi_{C_{e}}}{2},\frac{\Pi_{G_{e}}}{2},\frac{\Pi_{F_{e}}}{2} \end{array}\right\}$$

and

$$\begin{array}{c} C_{2}=\frac{\gamma^{2}}{2}+\frac{\Pi_{M_{e}}}{2}Tr\left({\left(M_{e}\right)}^{T}\left(M_{e}\right)\right)\;\;\;\;\;\;\;\;\;\;\;\;\;\;\;\;\;\;\\ +\frac{\Pi_{C_{e}}}{2}Tr\left({\left(C_{e}\right)}^{T}\left(C_{e}\right)\right)+\frac{\Pi_{G_{e}}}{2}Tr\left({\left(G_{e}\right)}^{T}\left(G_{e}\right)\right)\\ +\frac{\Pi_{F_{e}}}{2}Tr\left({\left(F_{e}\right)}^{T}\left(F_{e}\right)\right)+\frac{1}{2}{\tilde{\eta }}^{T}\left(K^{T}K\right)\tilde{\eta }\\ +Tr\left(\zeta {\tilde{\eta }}^{T}{\tilde{M}}_{e}+\alpha {\tilde{\eta }}^{T}{\tilde{C}}_{e}+\lambda {\tilde{\eta }}^{T}{\tilde{G}}_{e}+\lambda {\tilde{\eta }}^{T}{\tilde{F}}_{e}\right) \end{array}$$
In order to guarantee that 
$k_{2}>0$
, it is necessary to ensure that the controller parameters satisfy the conditions 
$(\lambda K-I_{n\times n})>0$
 and 
$\Pi_{{\left(.\right)}_{e}}>0$
.

In conclusion, we have the following theorem:

**Theorem** **4.**
*For the manipulator robotic system described by Equation (Equation 1), the implementation of the designed control scheme guarantees the stability of the system. This control scheme is based on the adaptive approximation technique Equation (Equation 41), augmented with the updated law Equation (Equation 42) and the state observer Equation (Equation 37). Additionally, all error signals, namely 
$\hat{\eta }$
, 
${\tilde{M}}_{e}$
, 
C˜e
, 
G˜e
, and 
${\tilde{F}}_{e}$
, are uniformly bounded (UBB) in the closed-loop form, in accordance with Lemma 2.*


**Proof.** The proof follows a similar approach to that of Theorem 3, and for the sake of brevity, the detailed proof will not be discussed here. □

## 4. Simulation and Comparative Analysis of Control Strategies

This section presents the results of the tests conducted to evaluate the effectiveness of the proposed controller through numerical simulations using Simulink (Matlab 2023a). The tests were performed on an exoskeleton robot system, as depicted in Figure 1, which consists of a 7-degree of freedom robot (7 DOFs) controlled by DC Maxon motors. The CAD model of the robot was created using SOLIDWORKS (version 2017) and imported into the virtual platform of *Gazebo*.

*Gazebo* served as a virtual environment where the proposed controller was implemented. To establish communication between the Simulink platform in Matlab and *Gazebo*, the Robot Operating System (ROS) was employed, as illustrated in Figure 1. ROS is a middleware operating system that enables the parallelization and coordination of multiple executables, known as “nodes”. In this context, ROS facilitates communication between Matlab (Simulink) and the exoskeleton robot. It allows executables to exchange information synchronously through topics or asynchronously through services. Topics enable subscription/publication, meaning that nodes can publish information on a topic, which can then be read by other nodes. On the other hand, services enable synchronous communication between two nodes.

The subsequent section presents the verification of the implemented control scheme on the exoskeleton device. The objective is to ensure that the controller is capable of accurately tracking the reference trajectories, even in the presence of unknown model dynamics and actuator parameters.

### 4.1. Implementation and Simulation of State-Feedback-Based Adaptive Modified Function Approximation Technique Control

In this subsection, simulation case studies were performed to validate the effectiveness of the designed controller (Equation (Equation 23)). In contrast to the experimental study that uses 7 DOFs, the simulation employs a rehabilitation task trajectory involving three degrees of freedom (3 DOFs). The reference trajectory for 
$q_{i}$
 was defined as follows: 
$q_{d}={\left[1.25-\left(\frac{7}{5}\right)e^{-t}+\left(\frac{7}{20}\right)e^{-4t},\;1.25+e^{-t}-\left(\frac{1}{4}\right)e^{-4t},\;1-\left(\frac{7}{5}\right)e^{-t}+\left(\frac{7}{20}\right)e^{-4t}\right]}^{T}$
. The initial values were set as: 
$q\left(0\right)={\left[0,\;1.25,\;0\right]}^{T}$
 rad and 
$\dot{q}\left(0\right)={\left[0,\;0,\;0\right]}^{T}$
 rad/s. The initial values of the control scheme were chosen as: 
$\left({\hat{M}}_{eii}=1\right)$
, 
$\left({\hat{C}}_{eii}=1\right)$
, 
$\left({\hat{G}}_{ei}=1\right)$
, and 
$\left({\hat{F}}_{ei}=1\right)$
. The parameters of the updating law in (Equation 34) were set to be: 
$\Lambda_{M_{e}}=10$
, 
$\Lambda_{C_{e}}=10$
, 
$\Lambda_{G_{e}}=10$
, and 
$\Lambda_{F_{e}}=10$
. In practice, 
$\Pi_{{\left(.\right)}_{e}}$
 can be represented as: 
$\Pi_{{\left(.\right)}_{e}}=\frac{1}{1+t^{2}}$
. The controller parameters were manually defined as: 
K=diag[30,80,30]
, 
β=0.5

Λ=2
, 
$\lambda_{1}=5.2$
, and 
$\lambda_{2}=3.5$
.

**Remark** **3.**
*All control gains presented in this paper were selected using a trial-and-error approach. We employed an iterative process to manually adjust the gains, aiming to achieve optimal system performance. This method allowed for a fine-tuned calibration of the control parameters based on practical experimentation and observation of the system’s response.*


### 4.2. Implementation and Simulation of Output-Feedback-Based Adaptive MFAT

In this subsection, simulation case studies were conducted to assess the effectiveness of the designed controller Equation (Equation 41). The performance of the second-order robust exact differentiator, defined by Equation (Equation 37), was evaluated using specific parameter values. The differentiator parameters were set as 
$\iota_{1}=3\iota^{1/3}$
, 
ι2=1.5ι1/3
, and 
ι3=1.1ι
, where 
$\iota ⩾\left|\dddot{q}\right|$
 and 
ι
 was fixed at 
ι=4.2
.

The updating law parameters in Equation (Equation 34) were chosen as follows: 
$\Lambda_{M_{e}}=0.1$
, 
ΛCe=0.1
, 
ΛGe=0.1
, and 
$\Lambda_{F_{e}}=0.1$
. Additionally, the controller parameters were manually set to 
K=diag[30,80,30]
, 
β=0.5
, 
Λ=2
, 
$\lambda_{1}=5.2$
, and 
$\lambda_{2}=3.5$
. These parameter values were selected to evaluate the performance and robustness of the controller in various scenarios.

### 4.3. Implementation of Conventional Function Approximation Technique Algorithm [18]

In this subsection, simulation case studies were conducted to assess the effectiveness of the conventional FAT controller [18]. The updating law parameters were set as follows: 
$\Lambda_{M_{e}}=0.1$
, 
ΛCe=0.1
, 
ΛGe=0.1
, and 
$\Lambda_{F_{e}}=0.1$
. The controller parameters were manually defined as 
K=diag[30, 80, 30]
 and 
Λ=2
. In contrast to the proposed adaptive approach, the initial weighting vectors of the FAT updated law were directly adopted from [21].

**Remark** **4.**
*To ensure better interpretability of the results, the conventional FAT approach control input [18] employed a modified 
sat(·)
 function instead of the 
sign(·)
 function. This adjustment was made to prevent excessive controller activity in the torque plot (Figure 13).*


Figure 2, Figure 3, Figure 4 and Figure 5 show the simulation results for the state-feedback adaptive modified function approximation technique (SFAT) using the control law in Equation (Equation 23). Figure 6, Figure 7, Figure 8, Figure 9 and Figure 10 display the results for the output feedback adaptive modified function approximation technique (OFAT) with the control law in Equation (Equation 41). Finally, Figure 11, Figure 12, Figure 13 and Figure 14 depict the outcomes for the conventional FAT approach [18].

Overall, the findings demonstrated smooth and effective operation, as seen in Figure 2, Figure 6, and Figure 11. Specifically, Figure 3 shows a substantial error reduction under the SFAT control law (Equation (Equation 23)), with fast convergence (less than 1 s). Furthermore, Figure 3 exhibits even faster convergence (less than 0.5 s) under the OFAT control law (Equation (Equation 41)). This is clearly an improvement over conventional FAT, where errors take around 2 s to converge, as shown in Figure 12.

Similar trends occurred for the estimated dynamic parameters (
$\hat{M}$
, 
C^
, 
G^
, and 
${\hat{F}}_{e}$
), quickly converging under SFAT (Equation 23) in Figure 5 and OFAT (Equation 41) in Figure 9, compared to FAT in Figure 14. Satisfactory control inputs (
τ
) are shown for the proposed approaches in Figure 4 and Figure 8, indicating effectiveness over the FAT controller [18] in Figure 13. Importantly, the SFAT and OFAT inputs were lower than the conventional FAT controller [18].

In summary, the manipulator demonstrated satisfactory performance under both the SFAT and OFAT controllers, despite a complete lack of knowledge of the dynamics, velocity (Figure 10), and disturbances. This confirms the strengths of the proposed approaches.

### 4.4. Comparative Study

A comparative study was performed to evaluate the developed controllers: SMFAT (Equation (Equation 23)), OMFAT (Equation (Equation 41)), and conventional FAT [18]. The analysis compared the Root-Mean-Squared (RMS) errors, sliding surfaces, and maximum controller input values. Table 1 summarizes the key performance metrics for each approach.

The results in Table 1 demonstrate that the SMFAT controller (Equation (Equation 23)) achieved satisfactory tracking performance with moderate control input, even without knowledge of the robot’s dynamics (*M*, *C*, and *G*) or actuator parameters (
$F_{e}$
). The OMFAT controller (Equation (Equation 41)) also had acceptable results, though its control input was relatively higher than both SMFAT (Equation (Equation 23)) and conventional FAT [18]. This increased input can be justified by the lack of prior knowledge of the manipulator dynamics (*M*, *C*, *G*, and 
$F_{e}$
) and unmeasured velocity state (
$\dot{q}$
) for OMFAT. In general, SMFAT delivered comparable tracking to OMFAT and FAT without requiring model knowledge or high control input. This confirmed the strengths of the proposed SMFAT method in handling unknown system dynamics and unmeasured states.

## 5. Experiments’ Results

To validate the proposed control scheme, real-time experiments were conducted on a 7-DOF exoskeleton robot called SREx. SREx is designed for rehabilitation applications enabling natural upper limb motion. It closely mimics human anatomy and can be worn to synchronize motions with the wearer during therapy. The design characteristics and mechanical properties of SREx were previously established in [39].

The SREx real-time system has three processing units (Figure 15). A PC runs the human–machine interface in LabVIEW 2017. A National Instruments PXI system executes the 500 μs top-level and 50 μs low-level control loops. Brushless DC motors (Maxon EC-45 and EC-90) combined with harmonic drives (120:1 ratio motors 1–2, 100:1 ratio motors 3–7) actuate the joints. Since velocity could not be directly measured, only the adaptive approximation technique (Equation (Equation 41)), updated law (Equation (Equation 42)), and state observer (Equation (Equation 37)) were implemented experimentally. The control architecture is shown in Figure 15.

The experimental results closely matched the simulation outcomes, demonstrating the effectiveness of the proposed technique. Similar to the simulations (Figure 6), the experiments showed smooth, accurate trajectory tracking (Figure 16). The tracking errors consistently decreased to near zero (Figure 17), indicating that the controller minimized deviations from the desired trajectory. The control input (
τ
) performed well under the control law (Equation (Equation 41)), as seen in Figure 19. Additionally, the higher-order sliding mode observer successfully estimated the velocity output state, evident from the close alignment between the desired and measured velocities (Figure 18). This matched the simulation results (Figure 10). In fact, these experimental results validated the proposed controller’s effectiveness in achieving satisfactory performance for the SREx exoskeleton manipulator with unknown dynamics, unmeasured velocity, and external disturbances. The experiments confirmed the simulations and demonstrated the feasibility of implementing the proposed approach on an actual robotic rehabilitation system. Figure 18 and Figure 19 are showed below

## 6. Conclusions

This study proposed an adaptive control strategy combining the modified function approximation technique (MFAT) and double-integral nonsingular terminal sliding mode control (DINTSMC) for robots with unknown dynamics using state and output feedback. Unlike conventional FAT, requiring basis functions, the modified FAT eliminates this need for model estimation. The integral components of DINTSMC enable continuous enhanced tracking, robustness, fast response, finite-time convergence, and reduced chattering.

The MFAT and higher-order sliding mode observer combination approximates the exoskeleton dynamics and accurately tracks the desired trajectories without velocity measurements. The simulations and experiments validated the proposed control scheme’s effectiveness. A comparative study of SMFAT, OFAT, and FAT revealed their relative strengths and weaknesses.

The proposed adaptive control strategy successfully addresses the limitations of previous techniques for rehabilitation robots with unknown nonlinear dynamics. The elimination of basis functions and integration of integral terminal sliding mode control contribute to improved performance. The comparative and experimental analyses confirmed the viability of implementing this approach on robotic rehabilitation systems.

Future work could develop an adaptive impedance force controller for real-time exoskeleton applications. This could further enhance stability across subjects and explore additional rehabilitation modes like active and active-assisted motion.

## Figures and Tables

**Figure 1 sensors-24-00489-f001:**
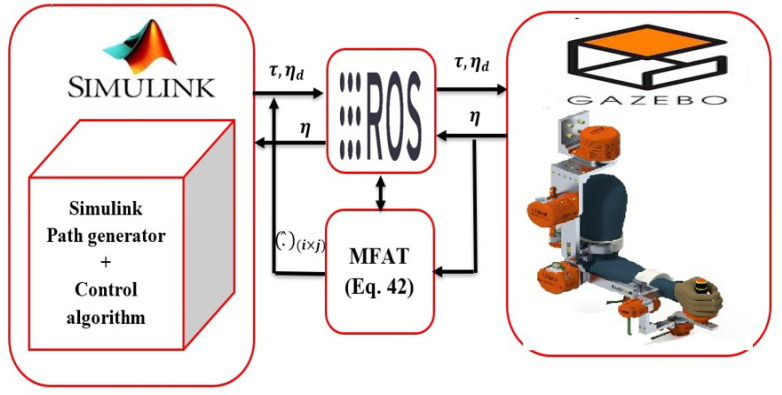
Simulation setup: control of joints in an exoskeleton manipulator robot (
$q_{1},q_{2}$
, and 
$q_{3}$
).

**Figure 2 sensors-24-00489-f002:**
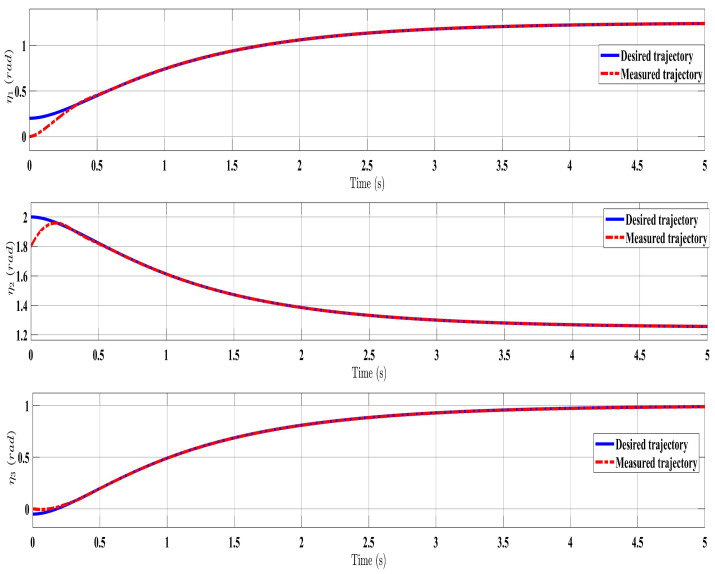
Comparing the measured and desired trajectories: control law (Equation (Equation 23)).

**Figure 3 sensors-24-00489-f003:**
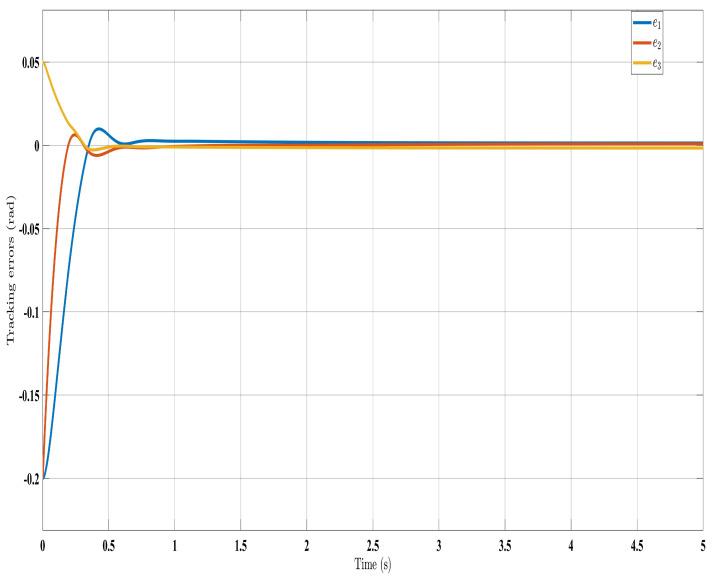
Evaluating tracking errors with control law (Equation (Equation 23)).

**Figure 4 sensors-24-00489-f004:**
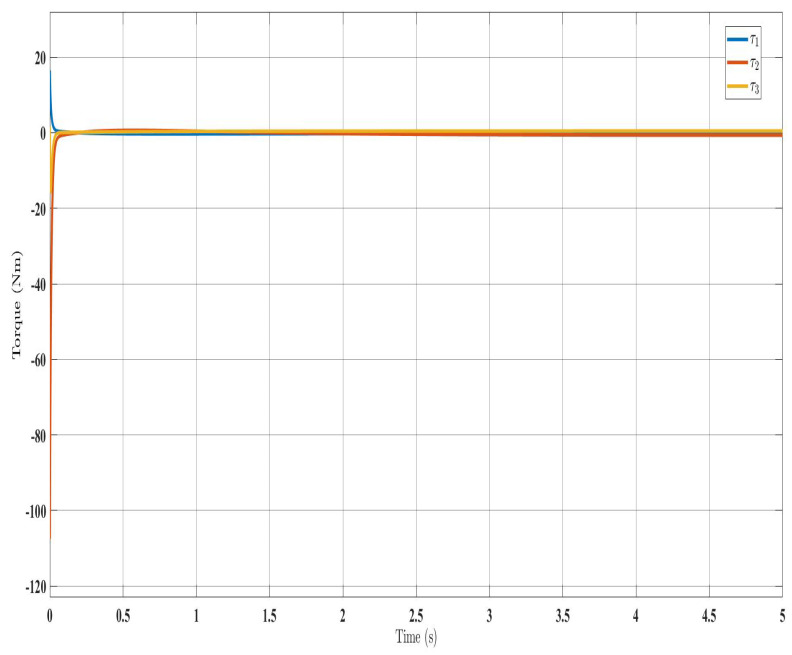
Torque input evolution with control law (Equation (Equation 23)).

**Figure 5 sensors-24-00489-f005:**
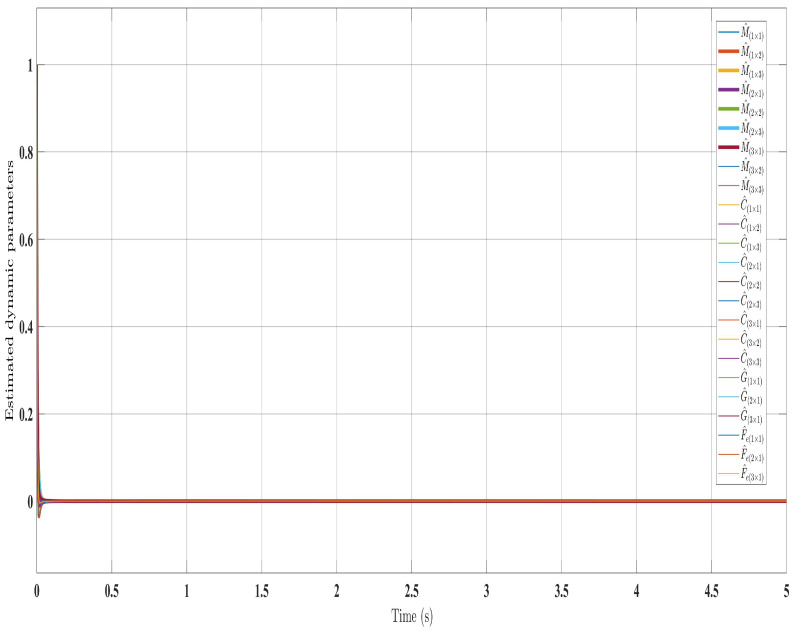
Estimated dynamic parameters using control law (Equation (Equation 23)).

**Figure 6 sensors-24-00489-f006:**
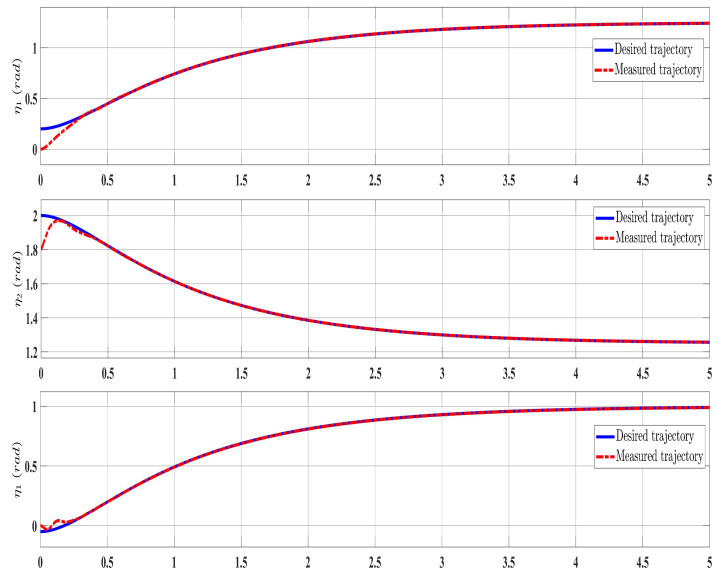
Comparing the measured and desired trajectories: control law (Equation (Equation 41)).

**Figure 7 sensors-24-00489-f007:**
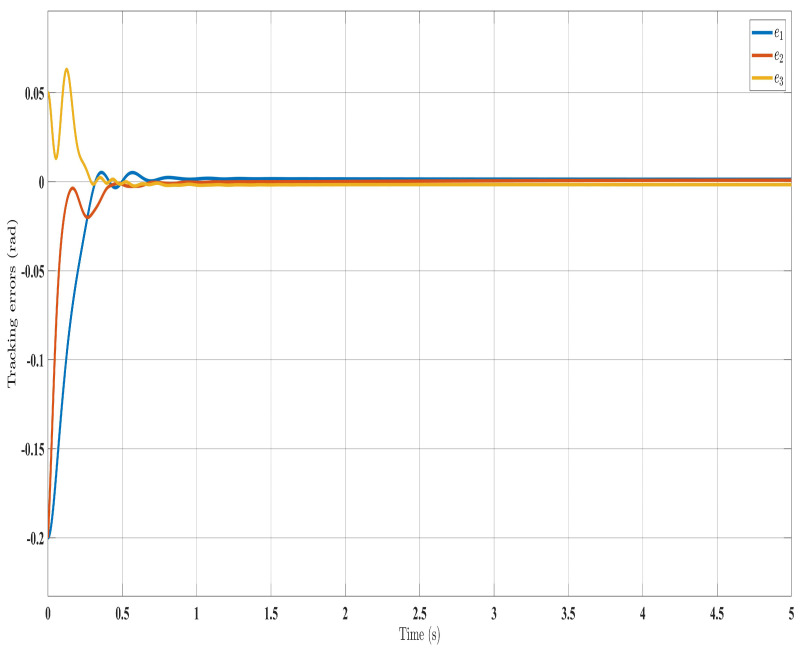
Tracking errors evolution with control law (Equation (Equation 41)).

**Figure 8 sensors-24-00489-f008:**
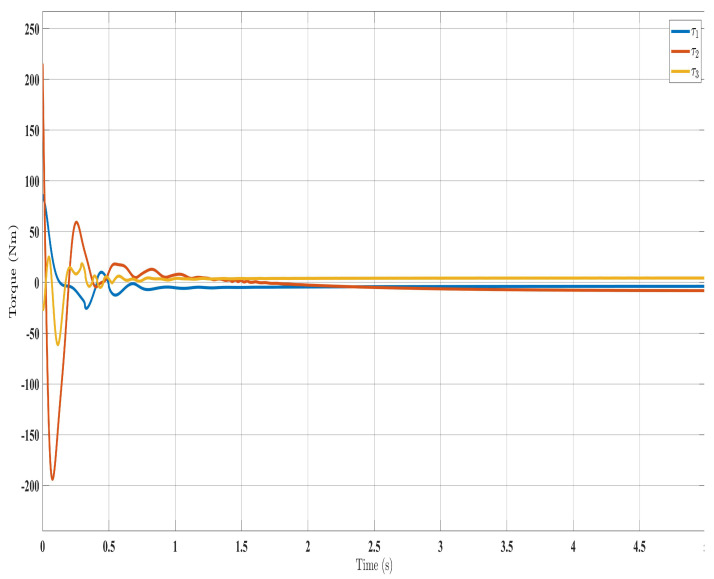
Torque input evolution with control law (Equation (Equation 41)).

**Figure 9 sensors-24-00489-f009:**
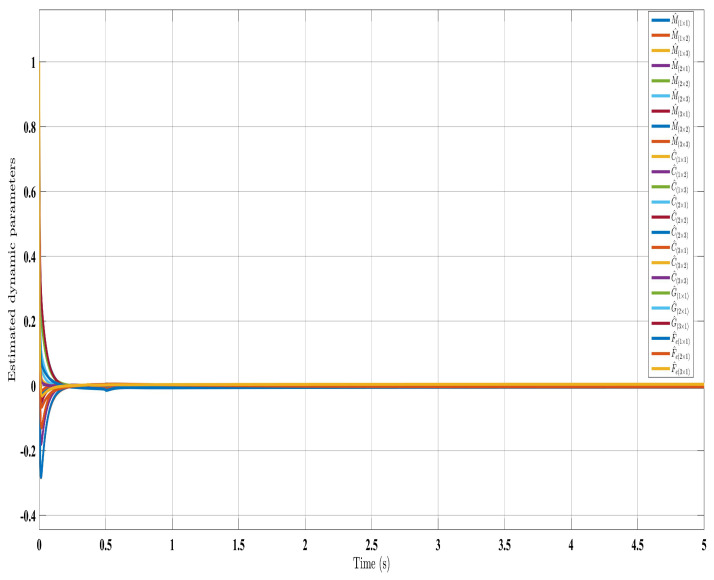
Estimated dynamic parameters with control law (Equation (Equation 41)).

**Figure 10 sensors-24-00489-f010:**
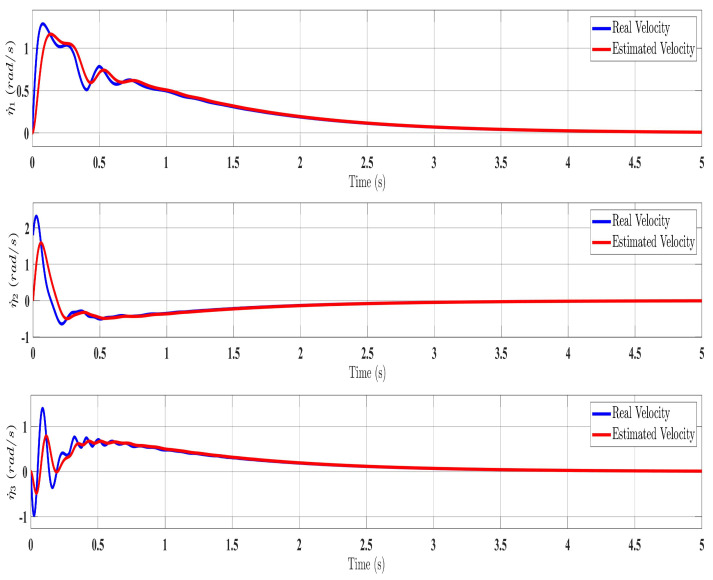
Estimated velocity with estimator law (Equation (Equation 37)).

**Figure 11 sensors-24-00489-f011:**
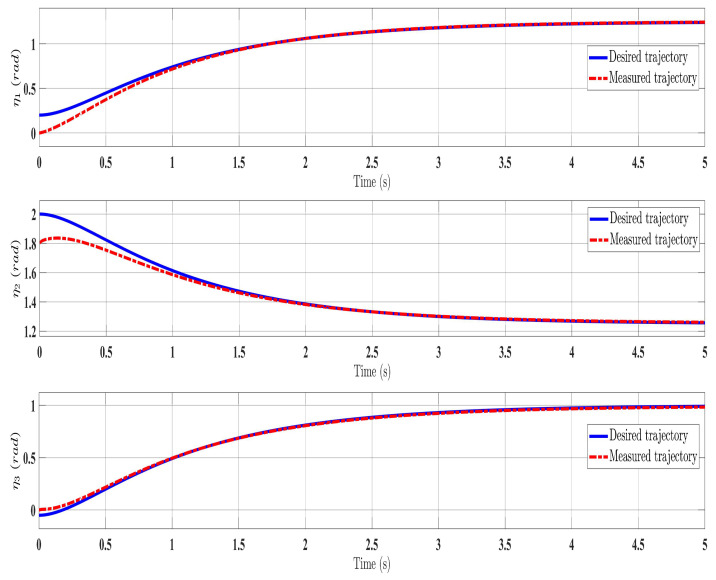
Comparing the measured and desired trajectories: control law [18].

**Figure 12 sensors-24-00489-f012:**
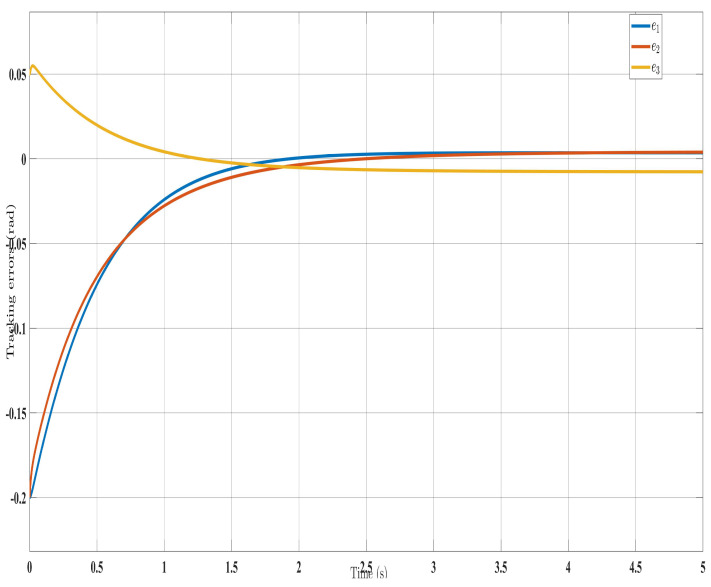
Tracking errors with control law [18].

**Figure 13 sensors-24-00489-f013:**
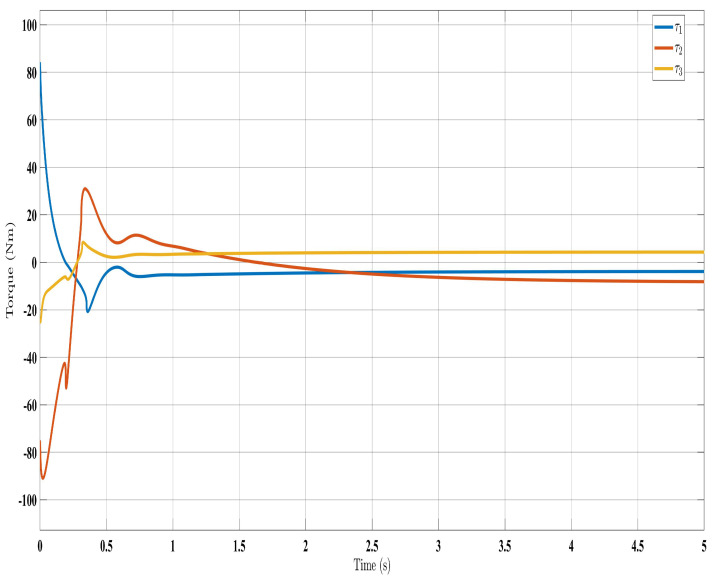
Torque input with control law [18].

**Figure 14 sensors-24-00489-f014:**
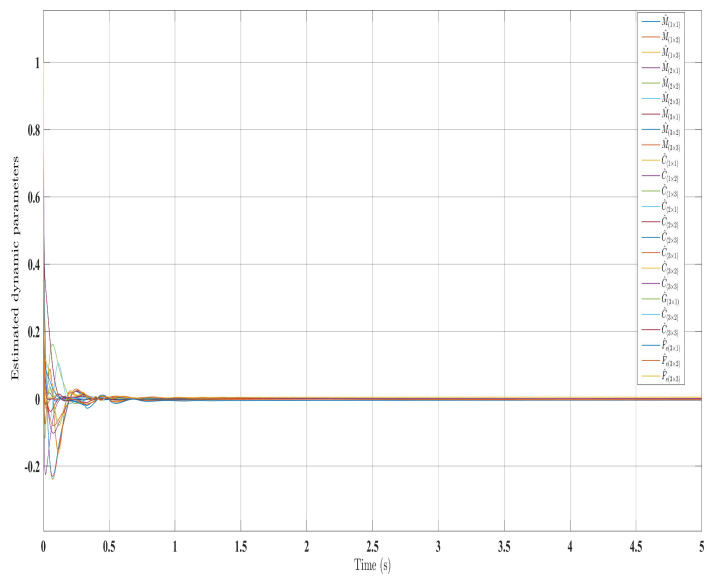
Estimated dynamic parameters with control law [18].

**Figure 15 sensors-24-00489-f015:**
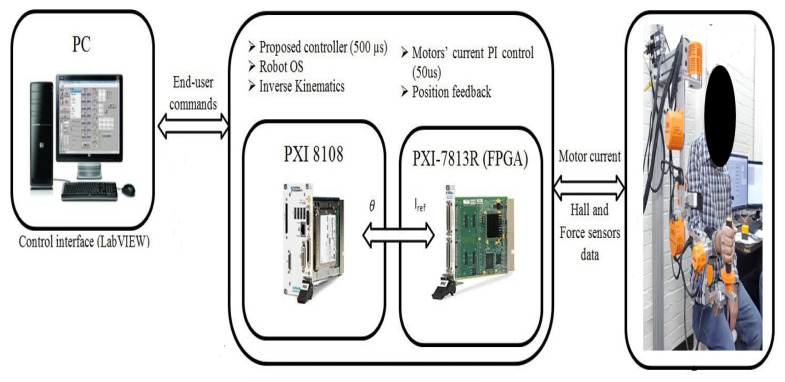
Experimental setup of the exoskeleton robot system: SREx.

**Figure 16 sensors-24-00489-f016:**
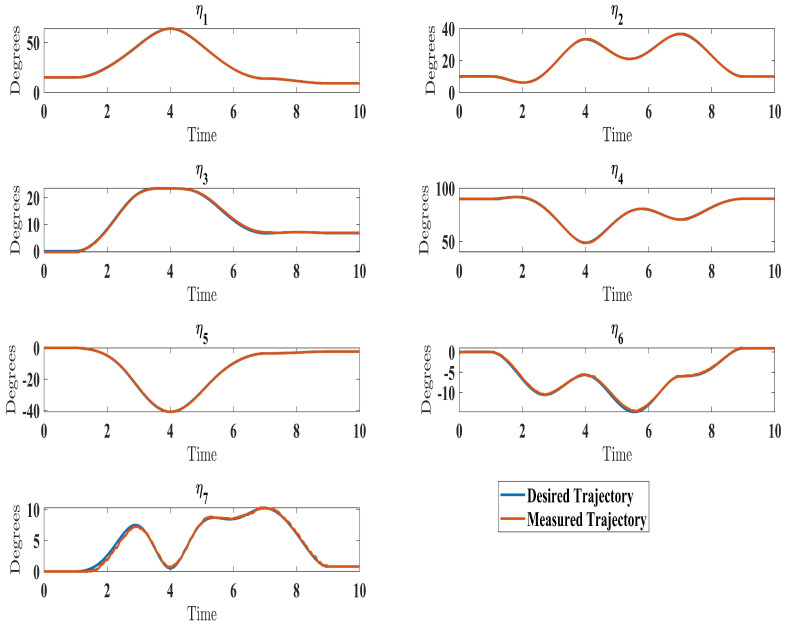
Comparison of measured and desired trajectories under control law (Equation (Equation 41)).

**Figure 17 sensors-24-00489-f017:**
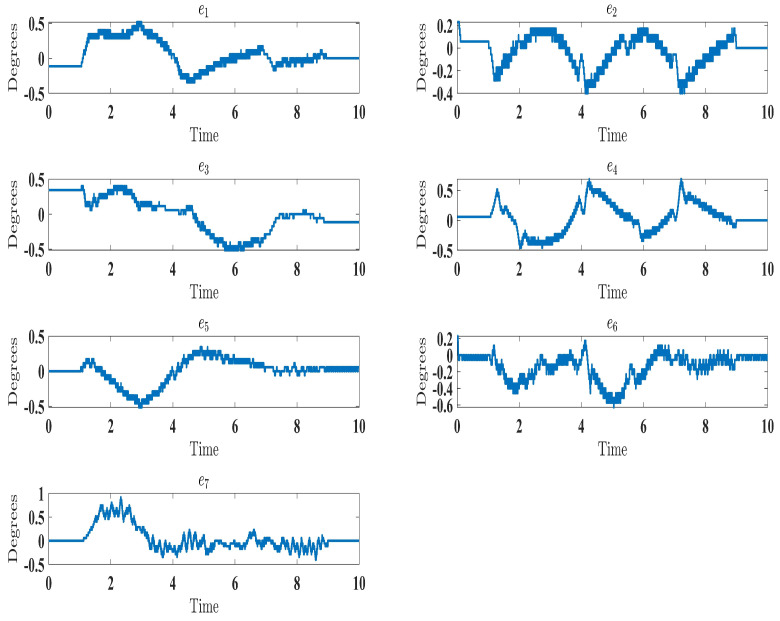
Evolution of tracking errors under control law (Equation (Equation 41)).

**Figure 18 sensors-24-00489-f018:**
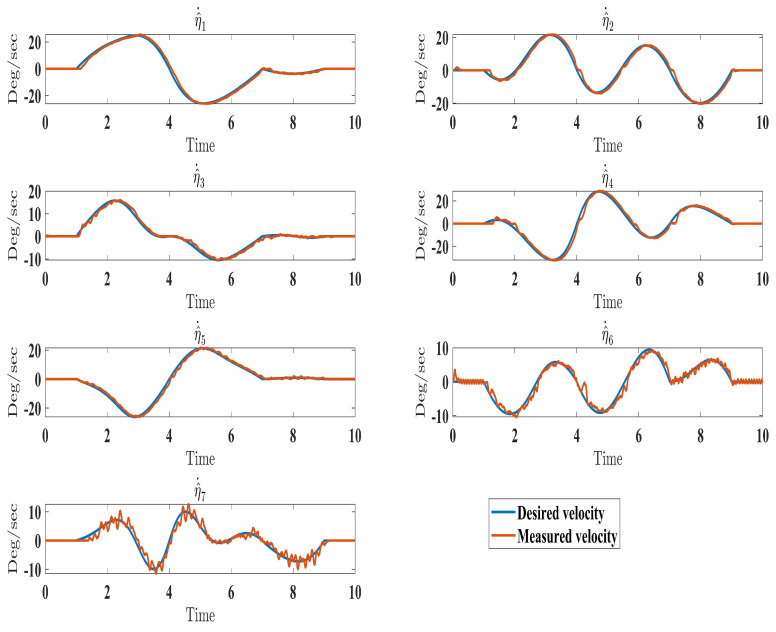
Evolution of estimated velocity under estimator law (Equation (Equation 37)).

**Figure 19 sensors-24-00489-f019:**
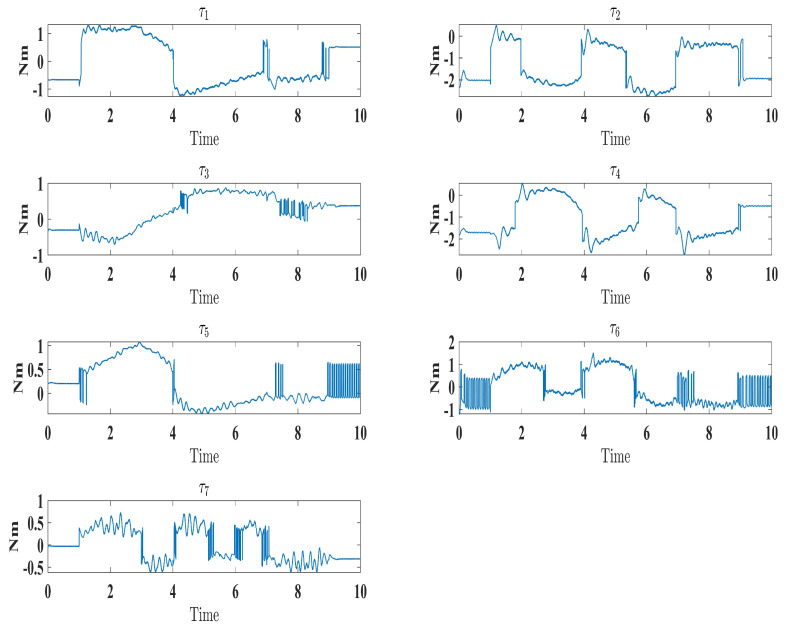
Evolution of torque input under control law (Equation (Equation 41)).

**Table 1 sensors-24-00489-t001:** Controller performance.

Regulator Variables	Control Input (Equation (Equation 23))	Control Input (Equation (Equation 41))	FAT Controller [18]
RMS ( $e_{1}$ )	0.0673	0.0886	0.1373
RMS ( $e_{2}$ )	0.0526	0.0722	0.1297
RMS ( $e_{3}$ )	0.0163	0.0207	0.0375
RMS ( $\eta_{1}$ )	0.1435	0.5052	0.7779
RMS ( $\eta_{2}$ )	0.4887	0.5309	0.7014
RMS ( $\eta_{3}$ )	0.1639	0.3492	0.3964
RMS ( $\tau_{1}$ )	5.6958	28.5111	24.9244
RMS ( $\tau_{2}$ )	38.0397	41.9870	57.5554
RMS ( $\tau_{3}$ )	4.6049	9.6346	11.4709

## Data Availability

The data that support the findings of this study are available from the corresponding author, Brahim Brahmi, upon reasonable request.

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
