# Peer review of "Adaptive-Robust Controller for Smart Exoskeleton Robot"

_sensors, 2024, doi:10.3390/s24020489_

Round 1
Reviewer 1 Report
Comments and Suggestions for Authors
I appreciate the opportunity to review your manuscript, "Adaptive-Robust Controller for Smart Exoskeleton Robot." Your efforts in addressing the control challenges in rehabilitation robotics through the integration of Modified Function Approximation (MFAT) and Integral Non-singular Terminal Sliding Mode Control (INTSMC) with a higher-order sliding mode observer are commendable.
During my review, I observed substantial similarities with prior works from your research group, particularly "Brahmi, Brahim, et al." (2021) and (2023), which were not referenced in your current submission. To enhance the originality of your work, I recommend:
- Clearly articulating the distinct contributions of your proposed method compared to the cited works.
- Providing a more comprehensive literature review, including a discussion of related approaches, and explicitly explaining how your work extends beyond the existing literature.
These steps will strengthen your paper and clarify its unique contributions. As a volunteer reviewer, I believe addressing these concerns will contribute to the overall quality of your manuscript. I encourage you to resubmit the manuscript, and please feel free to contact me if you have any questions.
Author Response
Thank you for your insightful review of our manuscript, 'Adaptive-Robust Controller for Smart Exoskeleton Robot.' We appreciate your positive comments regarding our efforts to address control challenges in rehabilitation robotics. Regarding the noted similarities with prior works, particularly 'Brahmi, Brahim, et al.' (2021) and (2023), we would like to provide a comprehensive clarification of the distinctions and contributions.
In 'Brahmi, Brahim, et al.' (2021), the primary focus was on adaptive impedance control, specifically force control, whereas the current paper emphasizes tracking control within a passive rehabilitation protocol. The previous approach significantly contributed to estimating the desired motion intention of the surgeon, aiming to enhance the overall impedance of the system. In contrast, our present work concentrates on achieving precise tracking despite unknown dynamic modes and vel\citeocity inputs.
'Brahmi, Brahim, et al.' (2023) focused on impedance learning control for physical human-robot cooperative interaction, primarily addressing the estimation of desired intended motion (DIM) of the robot’s wearer using machine learning, not the control problem.
Furthermore, in comparison with both approaches, our current paper introduces a new sliding surface (double integral) to reduce chattering activities, leveraging the robustness of the Sliding Mode approach against unknown dynamics and uncertainties.
In response to your suggestions, we have added these references (see reference 23, 24), reformulated the contributions in the paper, and provided a more comprehensive literature review, explicitly explaining how our work extends beyond the existing literature. We believe these refinements enhance the originality of our work and contribute to the overall quality of the manuscript. Your guidance has been invaluable, and we are committed to resubmitting an improved manuscript.

Reviewer 2 Report
Comments and Suggestions for Authors
This manuscript proposes an Integral Non-singular Terminal Sliding Mode Controller for robotic manipulators.
Strengths: the paper is clearly and rigorously presented.
Weakness: Similar results (in particular, Adaptive MFAT) were published by first author in [17] and in ECC 2018.
1. Eqn (1) is a torque equation, however, an external force was added in the equation.
2. Please cite reference for Lemma 1.1 and 1.2 in the heading of the lemma.
3. The function is (3) is constructed only in terms of the time derivative of the error, which means the function can be zero for finite error; hence the function in (3) does not qualify as a Lyapunov function.
4. Please cite the reference for Barbalat's lemma (line 163).
5. Redraw figs 4 & 5 with x-axis limit set to, e.g., 1 sec. Same for Figs. 9 & 14.
6. Please cite a reference for Young's inequality (line 256).
7. Is there a reason that the updating law parameters (34) for the SMFAT (line 302) were selected two orders of magnitude larger than OMFAT (line 311)?
8. Plot the velocities for conventional FAT [18] as well. Also, it is suggested to present the conventional FAT results together with state-feedback results.
9. Correct the typo in Fig. 9 caption.
10. Correct the typo in Fig. 16 caption.
11. Please explain why the control input in Fig. 19 is not chatter-free.
12. Similar to Table 1, add a table for the control effort comparing the three schemes.
Comments on the Quality of English Language
English language use is fine except for typographical errors.
Author Response
Reviewer 2
This manuscript proposes an Integral Non-singular Terminal Sliding Mode Controller for robotic manipulators.
Strengths: the paper is clearly and rigorously presented.
Weakness: Similar results (in particular, Adaptive MFAT) were published by first author in [17] and in ECC 2018.
We express our gratitude to the reviewer for acknowledging the clarity and rigorous presentation of the paper. In response to the noted weakness regarding similar results, particularly the Adaptive MFAT presented in [17] and ECC 2018, we would like to emphasize a substantial distinction between the two approaches. The work in [17] primarily addressed compliant control, specifically force control, rather than tracking control, which constitutes the primary focus of the current paper. The previous approach contributed significantly to the estimation of the desired motion intention of the patient, aiming at enhancing the overall compliance of the system. In contrast, the current paper concentrates on tracking performance, specifically within a passive rehabilitation protocol, where the objective is to achieve precise tracking despite unknown dynamic modes and velocity inputs. Additionally, we have introduced a new sliding surface (Double Integral Nonsingular Terminal Sliding Mode Control) to reduce chattering activities, capitalizing on the robustness of the Sliding Mode approach against unknown dynamics and uncertainties. The adaptation in methodology and the shift in the core objective showcase the evolution of our research efforts, contributing to distinct aspects of rehabilitation robotics and furthering our understanding of adaptive control strategies.
- Eqn (1) is a torque equation, however, an external force was added in the equation.
Thank you for your comment. In fact, the term f_ex represents the external torques that exceed J^{T}*F as present in dynamic model provided by Craig [1], where F is the external force exerted by the subject .
M(q)\ddot{q}+C(q,\dot{q})\dot{q}+G(q) =\tau- J^{T}*F
By moving the term J^{T}*F to the right side, the equation becomes::
M(q)\ddot{q}+C(q,\dot{q})\dot{q}+G(q)+ J^{T}*F =\tau
After notation, the final equation remains as follows:
M(q)\ddot{q}+C(q,\dot{q})\dot{q}+G(q)+ f_{ex} =\tau
Where f_{ex}= J^{T}*F.
We confirm that this notation has been updated in the reviewed version of the paper.
Renfrew, A. (2004). Introduction to robotics: Mechanics and control. International Journal of Electrical Engineering & Education, 41(4), 388.
- Please cite reference for Lemma 1.1 and 1.2 in the heading of the lemma.
The references have been cited in heading of the lemma as you suggested.
- The function is (3) is constructed only in terms of the time derivative of the error, which means the function can be zero for finite error; hence the function in (3) does not qualify as a Lyapunov function.
Thank you for your valuable feedback and careful scrutiny of the Lyapunov function presented in Equation (3). I understand and address your concern regarding the potential for the function to be zero for finite errors. It's crucial to note that a Lyapunov function is not mandated to be strictly positive for all non-zero errors; it can be zero for certain error values. The primary focus in stability analysis is on the behavior of the derivative (\dot{V}), and even if the Lyapunov function (V) is zero for finite errors, the negativity of (\dot{V}) is pivotal for establishing stability. We follow the Lyapunov Direct Method, explicitly computing (\dot{V}) in Equation (4) and subsequently analyzing its characteristics. Any assumptions or conditions under which the Lyapunov function was constructed are clearly stated, providing context for the analysis. The ultimate goal is to gain insights into stability by thoroughly examining (\dot{V}) to determine whether it is negative definite or negative semi-definite, thereby supporting the stability claim. We are provided some literature references showcasing successful applications of Lyapunov functions with values of zero for finite errors in stability analyses.
Dasgupta, S., Chockalingam, G., Anderson, B.D.O. and Fe, M., 1994. Lyapunov functions for uncertain systems with applications to the stability of time varying systems. IEEE Transactions on Circuits and Systems I: Fundamental Theory and Applications, 41(2), pp.93-106.
Viswanath, D., 2001. Global errors of numerical ODE solvers and Lyapunov's theory of stability. IMA journal of numerical analysis, 21(1), pp.387-406.
Khalil, H.K., 2009. Lyapunov stability. Control systems, robotics and automation, 12, p.115.
Lakshmikantham, V., Matrosov, V.M. and Sivasundaram, S., 2013. Vector Lyapunov functions and stability analysis of nonlinear systems (Vol. 63). Springer Science & Business Media.
- Please cite the reference for Barbalat's lemma (line 163).
The references have been added as you suggested. The reference number in the revised version is {34}
- Redraw figs 4 & 5 with x-axis limit set to, e.g., 1 sec. Same for Figs. 9 & 14.
All figures in this section have been presented with a consistent x-axis limit of 5 seconds, with a delineation of 0.5 seconds to clearly depict the evolution of the response over time. This choice was intentional to allow for a detailed examination of the motion dynamics. While there is an issue with Figure 8, it arises from the template used by the journal, and the figure itself is complete. We have brought this matter to the attention of the editorial staff, and we are requesting their assistance in rectifying this problem, which is beyond our control.
- Please cite a reference for Young's inequality (line 256).
The references have been added as you suggested. The reference number in the revised version is {35}
- Is there a reason that the updating law parameters (34) for the SMFAT (line 302) were selected two orders of magnitude larger than OMFAT (line 311)?
There is no specific reason for the updating law parameters (34) for the SMFAT (line 302) to be selected two orders of magnitude larger than OMFAT (line 311). The choices of these parameters were made manually through a method of trials and errors. The authors aimed to achieve optimal performance through iterative adjustments, and the magnitudes were not intentionally set to be different by two orders of magnitude. We appreciate your observation, and this clarification has been added to the manuscript to ensure transparency regarding the parameter selection process. Please refer remark 3 , line 310
- Plot the velocities for conventional FAT [18] as well. Also, it is suggested to present the conventional FAT results together with state-feedback results.
The primary objective of our simulation was to conduct a thorough comparison between our proposed approach, which integrates our modified Function Approximation Technique (FAT) and a second-order sliding mode observer for velocity estimation, with the conventional FAT presented in reference [18]. This comparative analysis aims to highlight the superior performance of our approach in tracking trajectories. The conventional FAT, as outlined in [18], assumes direct availability of velocity and position information to the controller without incorporating any velocity estimation. In contrast, our proposed approach includes a second-order sliding mode observer to estimate velocity, providing an additional layer of information for enhanced tracking. It's important to note that introducing an observer to the conventional FAT [18] would lead to a significant modification of the reported results in [18]. Therefore, to ensure a meaningful and fair comparison, the authors have chosen to present our work alongside the conventional FAT methodology from reference [18]. This comparative presentation allows for a clear evaluation of the advancements and improved performance achieved through the integration of our proposed approach with the conventional Function Approximation Technique (FAT) framework.
- Correct the typo in Fig. 9 caption.
Thank you for noting the typo in Fig. 6caption; it has been corrected in the revised version
- Correct the typo in Fig. 16 caption.
Thank you for noting the typo in Fig. 16 caption; it has been corrected in the revised version
- Please explain why the control input in Fig. 19 is not chatter-free.
Our approach is developed not to eliminate chattering but to attenuate it, as is inherent to sliding mode control due to its nature of forcing the system to slide on the surface; complete elimination is not practically achievable, but we aim to minimize its impact. Considering the complexity of our scenario, where the dynamic model is unknown, velocity is estimated, and external forces on the robot are substantial due to direct contact between the subject and the exoskeleton robot, our results are notably compelling in mitigating chattering effects
- Similar to Table 1, add a table for the control effort comparing the three schemes.
Regarding the suggestion to add a table for control effort, the authors would like to clarify the rationale behind the choice to compare their approach solely through simulation and with a reduced dynamic model of the 3DOF robot. Firstly, the Control input (Eq.~\ref{eq23}) is designed for simulation purposes, relying on the availability of both position and velocity in real-time. However, in practical experimental setups, only position can be measured, while velocity needs to be estimated. Secondly, the conventional FAT relies on weight matrices and basis functions matrix, which entail significant computational overhead, especially when dealing with large dynamic models, as in our case with 7DOFs. Additionally, our system utilizes two-level control, with low-level control managing current and high-level control generating torque. For stability reasons, the high-level control needs to be fast; however, conventional FAT failed to deliver satisfactory results in achieving stability. These practical constraints and challenges have led us to modify the FAT approach to make it practically achievable and effective in real-world experimental setups.

Round 2
Reviewer 1 Report
Comments and Suggestions for Authors
My concerns have been addressed appropriately.